# Control of adaptive action selection by secondary motor cortex during flexible visual categorization

Tian-Yi Wang[1,2], Jing Liu[1,2], Haishan Yao[1,3]*

[1]Institute of Neuroscience, State Key Laboratory of Neuroscience, CAS Center for Excellence in Brain Science and Intelligence Technology, Chinese Academy of Sciences, Shanghai, China; [2]University of Chinese Academy of Sciences, Beijing, China; [3]Shanghai Center for Brain Science and Brain-Inspired Intelligence Technology, Shanghai, China

**Abstract** Adaptive action selection during stimulus categorization is an important feature of flexible behavior. To examine neural mechanism underlying this process, we trained mice to categorize the spatial frequencies of visual stimuli according to a boundary that changed between blocks of trials in a session. Using a model with a dynamic decision criterion, we found that sensory history was important for adaptive action selection after the switch of boundary. Bilateral inactivation of the secondary motor cortex (M2) impaired adaptive action selection by reducing the behavioral influence of sensory history. Electrophysiological recordings showed that M2 neurons carried more information about upcoming choice and previous sensory stimuli when sensorimotor association was being remapped than when it was stable. Thus, M2 causally contributes to flexible action selection during stimulus categorization, with the representations of upcoming choice and sensory history regulated by the demand to remap stimulus-action association.

**\*For correspondence:**
haishanyao@ion.ac.cn

**Competing interests:** The authors declare that no competing interests exist.

## Introduction

Flexible action selection allows the animal to adapt to changes in the relations between stimulus, response and outcomes (*Gold and Stocker, 2017*; *Ragozzino, 2007*; *Wise and Murray, 2000*). Specifically, during flexible stimulus categorization, changes in the categorical boundary requires a remapping between particular sensory stimuli and behavioral responses. For instance, a line segment can be classified as short or long, depending on the criterion length (*Grinband et al., 2006*). Rodents as well as primates demonstrate the ability to flexibly adjust stimulus-action association after the switch of categorical boundary (*Ferrera et al., 2009*; *Grinband et al., 2006*; *Jaramillo et al., 2014*; *Mendez et al., 2011*). In flexible sound categorization task, the performance of rodents could be accounted for by a distance-to-boundary model, in which the animal compared the current stimulus with an internal categorical boundary (*Jaramillo et al., 2014*). Previous works showed that perceptual decisions are influenced not only by sensory input but also trial history, which includes history of past stimuli and choice outcome (*Akrami et al., 2018*; *Busse et al., 2011*; *Hwang et al., 2017*; *Lak et al., 2020*; *Thompson et al., 2016*). During flexible stimulus categorization task, it is unclear how different aspects of trial history influence the internal categorical boundary and the action selection after boundary switch. Applying computational models incorporating history factors could reveal the strategies that underlie the flexible response (*Churchland and Kiani, 2016*).

Neural representation of stimulus category has been found in lateral intraparietal cortex (*Freedman and Assad, 2006*), prefrontal cortex (*Freedman et al., 2001*) and medial premotor cortex (*Romo et al., 1997*). Neural correlates of flexible stimulus categorization are found in monkey pre-supplementary motor cortex (*Mendoza et al., 2018*) and the frontal eye field (*Ferrera et al.,*

*2009*). Neurons in monkey prefrontal and anterior cingulate cortex also showed dynamic task selectivity in task switching that requires flexible adjustment of behavior (*Johnston et al., 2007*). However, the causal role of frontal and premotor regions in the performance of flexible stimulus categorization remains to be investigated.

The secondary motor cortex (M2) in rodents is a homolog of the premotor cortex, supplementary motor area, or frontal eye field in monkey (*Barthas and Kwan, 2017*; *Reep et al., 1987*; *Svoboda and Li, 2018*; *Zingg et al., 2014*). M2 plays a critical role in the flexible control of voluntary action (*Barthas and Kwan, 2017*; *Ebbesen et al., 2018*). Removal or inactivation of M2 caused deficits in cue-guided actions (*Barthas and Kwan, 2017*; *Erlich et al., 2015*; *Passingham et al., 1988*), and an increase in errors during behavioral switch from nonconditional responding to cue-guided actions (*Siniscalchi et al., 2016*). M2 neurons exhibited choice-related activity, which is earlier than that in other frontal cortical regions (*Sul et al., 2011*). Neural signals in M2 also conveyed information about past choice and outcome (*Hattori et al., 2019*; *Jiang et al., 2019*; *Scott et al., 2017*; *Siniscalchi et al., 2019*; *Sul et al., 2011*; *Yuan et al., 2015*). The findings in these previous studies raise the possibility that M2 may be important for adaptive action selection during flexible stimulus categorization. Furthermore, it would be interesting to elucidate whether the choice- and history-related signals in M2 are modulated by the task demand to remap stimulus-action association.

Here, we combined behavioral modeling, chemogenetic manipulation and extracellular recording to explore the mechanism underlying adaptive action selection. Freely-moving mice categorized the spatial frequencies (SFs) of gratings as low or high according to a boundary that shifted between a lower and a higher frequency across blocks of trials. For the reversing stimulus whose frequency lied between the two boundaries, the mice should reverse their choice as the boundary switched. Using a behavioral model in which the decision criterion (DC) was updated according to the history of action outcome and sensory stimuli, we found that sensory history was important for correct adjustment of DC during the switching period. Bilateral inactivation of M2 impaired the performance of reversing response by reducing the mice's ability to update DC according to sensory history, without affecting the performance in trials when the stimulus-action association was stable. Furthermore, M2 neurons encoded the upcoming choice and stimulus in previous trial more accurately during the switching than the stable period. Together, the results suggest that during stimulus categorization, M2 is involved in flexible adjustment of stimulus-action association in the face of boundary change.

## Results

### Flexible visual categorization task for freely-moving mice

For the visual system, the perceived size (or SF) of a visual object changes with viewing distance, and categorization of a visual stimulus as low or high SF may be adaptive to the change of viewing distance. In our study, we trained freely-moving mice to categorize visual stimuli as low or high SFs using a two-alternative forced-choice paradigm. Similar to the auditory flexible categorization task described in a previous study (*Jaramillo et al., 2014*), we changed the categorization boundary in different blocks of trials. The mouse poked its nose in the central port of a behavioral chamber to initiate a trial (*Long et al., 2015*). A static grating was presented on the screen, and the mouse was required to maintain its head in the central port until a "Go' signal, a light-emitting diode, was turned on to indicate that the mouse could choose one of the two side ports (*Figure 1A*). The rewarded side port was on the left (or right) if the grating SF was lower (or higher) than a categorical boundary, which was not cued but learned by the mouse through trial and error. The visual stimuli consisted of 7 SFs that were logarithmically equally-spaced (0.03, 0.044, 0.065, 0.095, 0.139, 0.204 and 0.3 cycles/°). Within a session, the categorical boundary shifted between a lower and a higher SF several times without a warning cue (*Figure 1B*). For the low-boundary block, the optimal decision boundary was located between 0.065 and 0.095 cycles/°, and gratings at 0.03 and 0.095 cycles/° (SF1 and SF4 in *Figure 1B*) were frequently presented in 90% of trials. For the high-boundary block, the optimal decision boundary was located between 0.095 and 0.139 cycles/°, and gratings at 0.095 and 0.3 cycles/° (SF4 and SF7 in *Figure 1B*) were frequently presented in 90% of trials. As a result, the stimulus statistics differed between the low-boundary and high-boundary blocks. In each session, each block consisted of at least 60 trials, and the categorical boundary switched once the performance for the reversing stimulus (0.095 cycles/°, SF4 in *Figure 1B*) reached

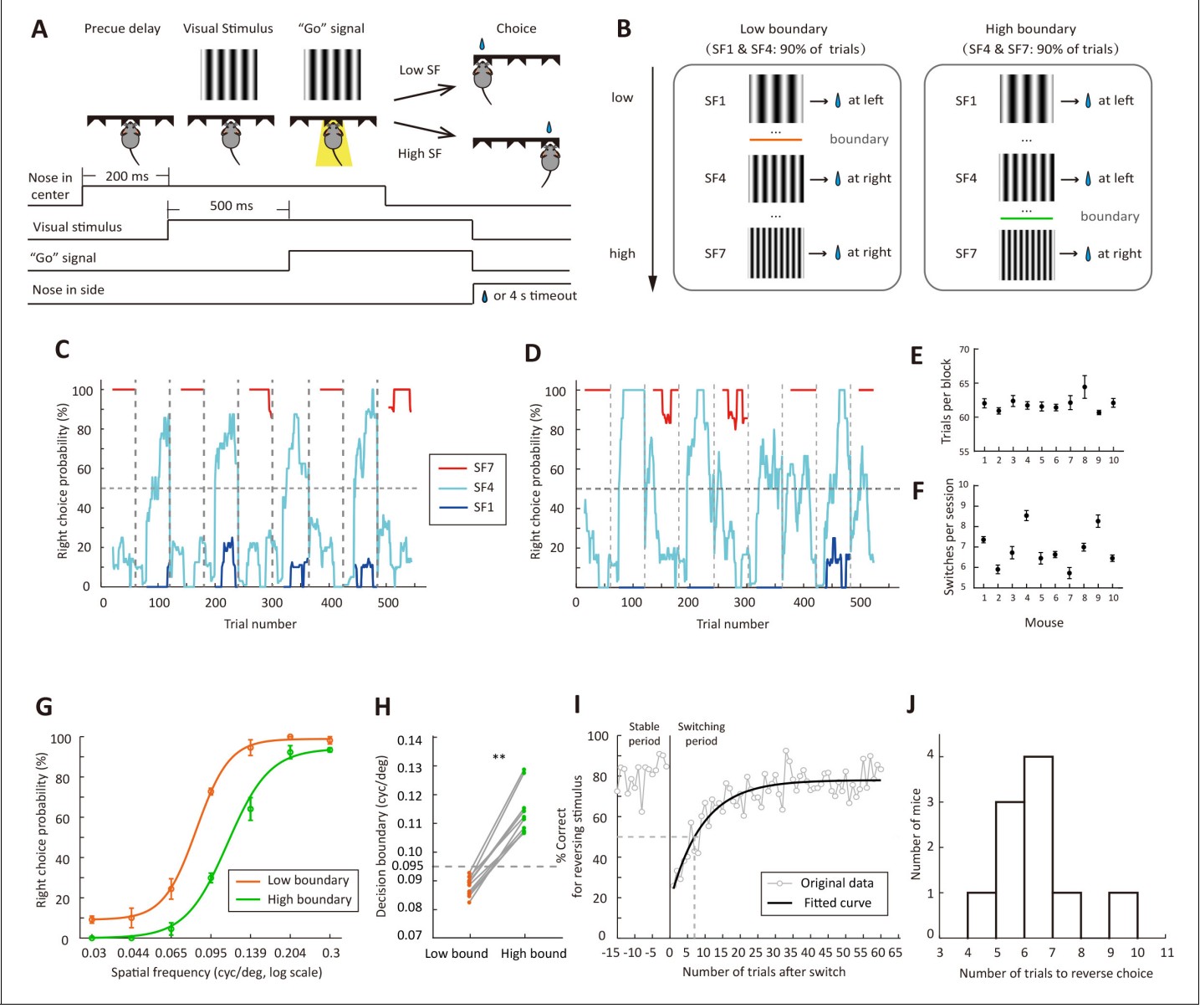

**Figure 1.** Flexible visual categorization task. (**A**) Schematic of the task and timing of behavioral events. (**B**) Visual stimuli and categorical boundaries. Each session consisted of alternating low-boundary and high-boundary blocks, the order of which was counterbalanced across sessions. For low-boundary blocks, SF1 and SF4 were presented for 90% of trials; for high-boundary blocks, SF4 and SF7 were presented for 90% of trials. (**C and D**) Performance of one example mouse for SF1 (0.03 cycles/°), SF4 (0.095 cycles/°) and SF7 (0.3 cycles/°) in two different sessions. For each trial, the probability of right choice was computed over the previous 15 trials. H, high-boundary block; L, low-boundary block. (**E**) Number of trials per block for each mouse. (**F**) Number of switches per session for each mouse. (**G**) Psychometric curves from low-boundary and high-boundary blocks for an example mouse (data from 11 sessions). (**H**) Comparison of internal decision boundary between blocks for a population of mice. p=2 × $10^{-3}$, n = 10 mice, Wilcoxon signed rank test. (**I**) Performance for the reversing stimulus before and after the boundary switch for an example mouse. The curve is an exponential fit of the data after the boundary switch. (**J**) Distribution of the number of trials to reverse choice, which was the number of trials for the correct rate of reversing stimulus to reach 50% after the boundary switch. n = 10 mice. Error bar,± SEM. See *Figure 1—source data 1–2* for complete statistics.

The online version of this article includes the following source data for figure 1:

**Source data 1.** Number of trials per block, number of switches per session and number of trials to reverse choice for a population of mice.
**Source data 2.** Psychometric curves and decision boundary for low-boundary and high-boundary blocks.

70% over the last 10 reversing-stimulus trials. After the boundary changed, the mouse was required to reverse its choice for the reversing stimulus (*Figure 1C and D*). Across all sessions from 10 mice (11 sessions/mouse), the number of trials in each block was 61.9 ± 0.33 (mean ± SEM) and the number of switches in a session was 6.91 ± 0.29 (mean ± SEM) (*Figure 1E and F*).

Although only the action for the reversing stimulus was required to change, we found that the performance for both the reversing and the non-reversing stimuli was influenced by the change of categorical boundary (*Figure 1G*), consistent with that observed in auditory flexible categorization task (*Jaramillo et al., 2014*). The subjective categorical boundary estimated from the psychometric curve was significantly lower in the low-boundary than in the high-boundary block (p=2 × $10^{-3}$, n = 10 mice, Wilcoxon signed rank test, *Figure 1H*), suggesting that the mice adapted their DC to the boundary contingency. To examine how fast the mice adapted to the boundary change, we used all blocks in all sessions to compute correct rate for the reversing stimulus in each trial. The performance in the first 60 trials after boundary switch was fitted with an exponential function (*Figure 1I*). The number of trials to reverse choice, which was the number of trials needed to cross the 50% correct rate of the fitted curve, was 6.41 ± 0.42 (mean ± SEM, n = 10 mice, *Figure 1J*). In the following analysis, we defined the last 15 trials before boundary switch as the stable period, and the first 15 trials after boundary switch as the switching period.

## Behavioral strategies revealed by computational modeling

After the boundary switch, the animals may update their stimulus-action association according to the outcome of response to the reversing stimulus and/or the appearance of non-reversing stimulus frequently presented in that block (*Jaramillo and Zador, 2014*). To examine the behavioral strategies of mice in trials before and after the boundary switch, we designed a logistic regression model with a dynamic DC to fit the mouse's choice on each trial. In this model, the probability $p$ of choosing right in current trial $t$ is a logistic function of a decision variable $z$, which is the difference between the stimulus S weighted by γ1 (γ1 > 0) and the subjective DC of the animal (*Figure 2A*, see Materials and methods). The SF of stimulus S is normalized between −1 and 1, in which negative and positive values indicate SFs lower and higher than the SF of the reversing stimulus, respectively. The DC is updated on a trial-by-trial basis according to choice outcome and sensory history (*Figure 2A*). Following a trial of reversing stimulus, the DC is updated according to the outcome of previous choice, with the effect of rewarded and unrewarded choices modeled by the parameters α1 and α2, respectively. Considering that DC may exhibit a drift tendency independent of trial history, a parameter β (β <1) was included to describe how fast DC drifts toward zero (0 < β <1) or away from zero (β <0). For α1 (α2) with a positive value, rewarded (unrewarded) choices would introduce a bias towards (away from) the previously chosen side, mimicking a win-stay (lose-shift) strategy. Similarly, α1 (α2) with a negative value represents a win-shift (lose-stay) strategy. Following a trial of non-reversing stimulus, the DC is updated according to the sensory history, with the previous stimulus weighted by the parameter γ2. For γ2 with a positive value, a previously experienced low-frequency (high-frequency) stimulus would cause a shift of DC towards low (high) frequency, leading to a right (left) choice bias.

We modeled the choices of individual trials for each mouse by combining data across multiple sessions. We first used the model to fit choices in the stable and switching periods separately so that we could compare the strategies between the two periods (*Figure 2B and C*). For both the stable and switching periods, γ1 was significantly larger than zero (p=2 × $10^{-3}$, n = 10, Wilcoxon signed rank test), indicating that current sensory information was used to form decision and guide action selection in both periods. We next examined the values of α1, α2, γ2 and β, which are relevant to the update of DC (*Figure 2B−D*). The reward history parameter α1 was not significantly different from zero in either the stable (p=0.23) or the switching period (p=0.49, Wilcoxon signed rank test). The non-reward history parameter α2 was significantly larger than zero in the stable period (p=9.8 × $10^{-3}$, Wilcoxon signed rank test), but not significantly different from zero in the switching period (p=0.7, Wilcoxon signed rank test), suggesting that the mice adopted a lose-shift strategy only in the stable period. The sensory history parameter γ2 was significantly larger than zero in both the stable and the switching periods (p=0.037 and p=2 × $10^{-3}$, Wilcoxon signed rank test), suggesting that the mice could optimally use the sensory history of non-reversing stimuli to guide choice in both periods. The DC-drift parameter β was significantly larger than zero in the switching period (p=2 × $10^{-3}$) but not in the stable period (p=0.56, Wilcoxon signed rank test). This demonstrates

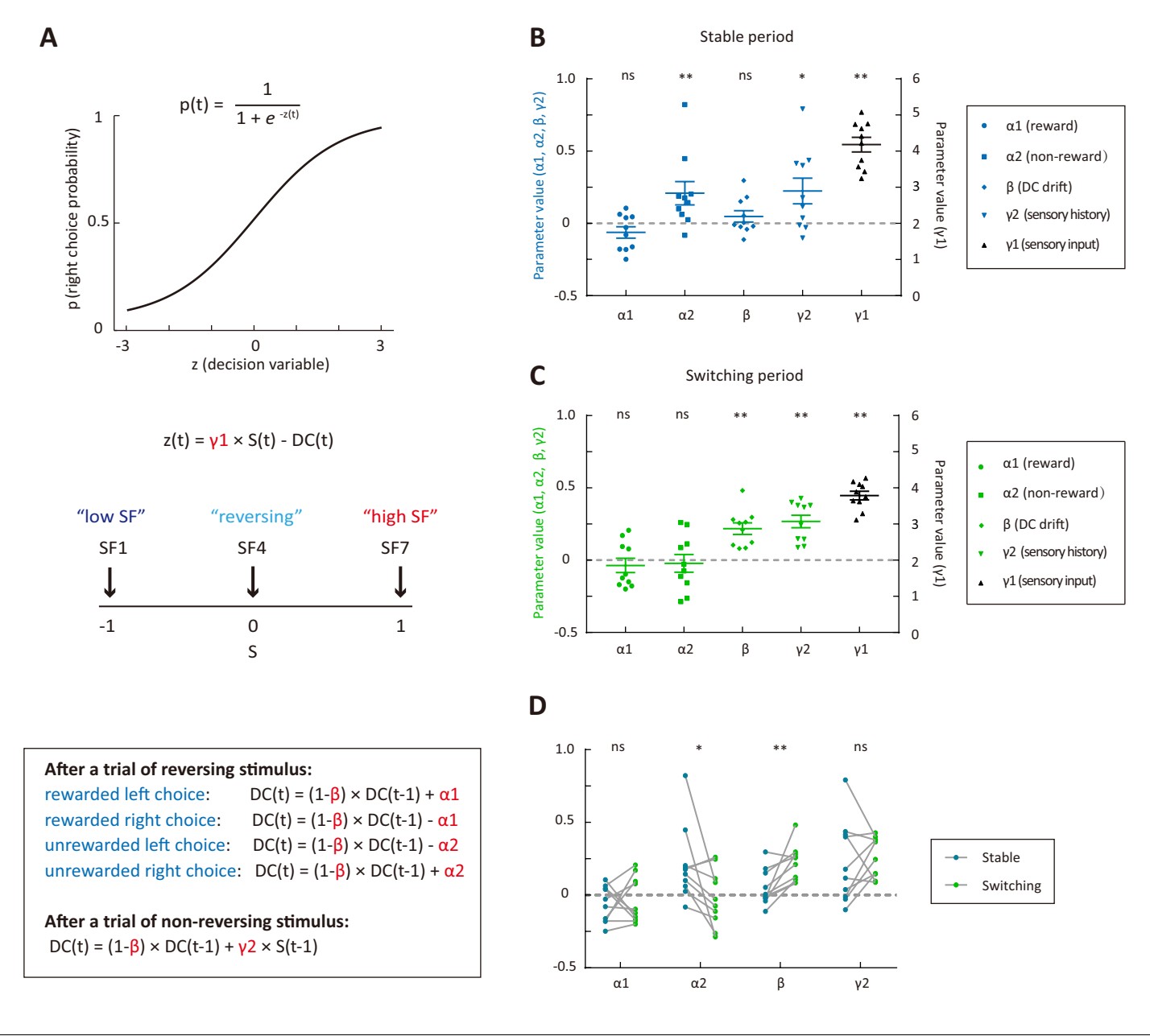

**Figure 2.** Behavioral strategies revealed by computational modelling. (**A**) Schematic of the dynamic DC model, in which the DC was updated on a trial-by-trial basis according to choice outcome history ($\alpha1$ and $\alpha2$) or sensory history ($\gamma2$). (**B**) Model parameters for the stable period. (**C**) Model parameters for the switching period. (**D**) Comparison of parameters important for the update of DC between the stable and the switching periods. *$p<0.05$, **$p<0.01$, n = 10 mice, Wilcoxon signed rank test. See *Figure 2—source data 1* for complete statistics.

The online version of this article includes the following source data for figure 2:

**Source data 1.** Parameters of the dynamic-DC model fitted to choices in the stable and switching periods separately.

that the DC tended to drift towards zero in the switching period, reflecting the tendency of mice to abandon pervious decision criterion.

To evaluate whether each model parameter is necessary, we further built reduced model variants (model 2: $\alpha1 = 0$; model 3: $\alpha2 = 0$; model 4: $\alpha1 = 0$ and $\alpha2 = 0$; model 5: $\beta = 0$; model 6: $\gamma1 = 1$; model 7: $\gamma2 = 0$) (*Figure 3*) to compare with the full model (model 1). For the stable period, only model 6 ($\gamma1 = 1$) showed a significantly lower cross-validated (CV) likelihood as compared to that of the full model ($p=2 \times 10^{-3}$, Wilcoxon signed rank test, *Figure 3A*), indicating that $\gamma1$ (weight of

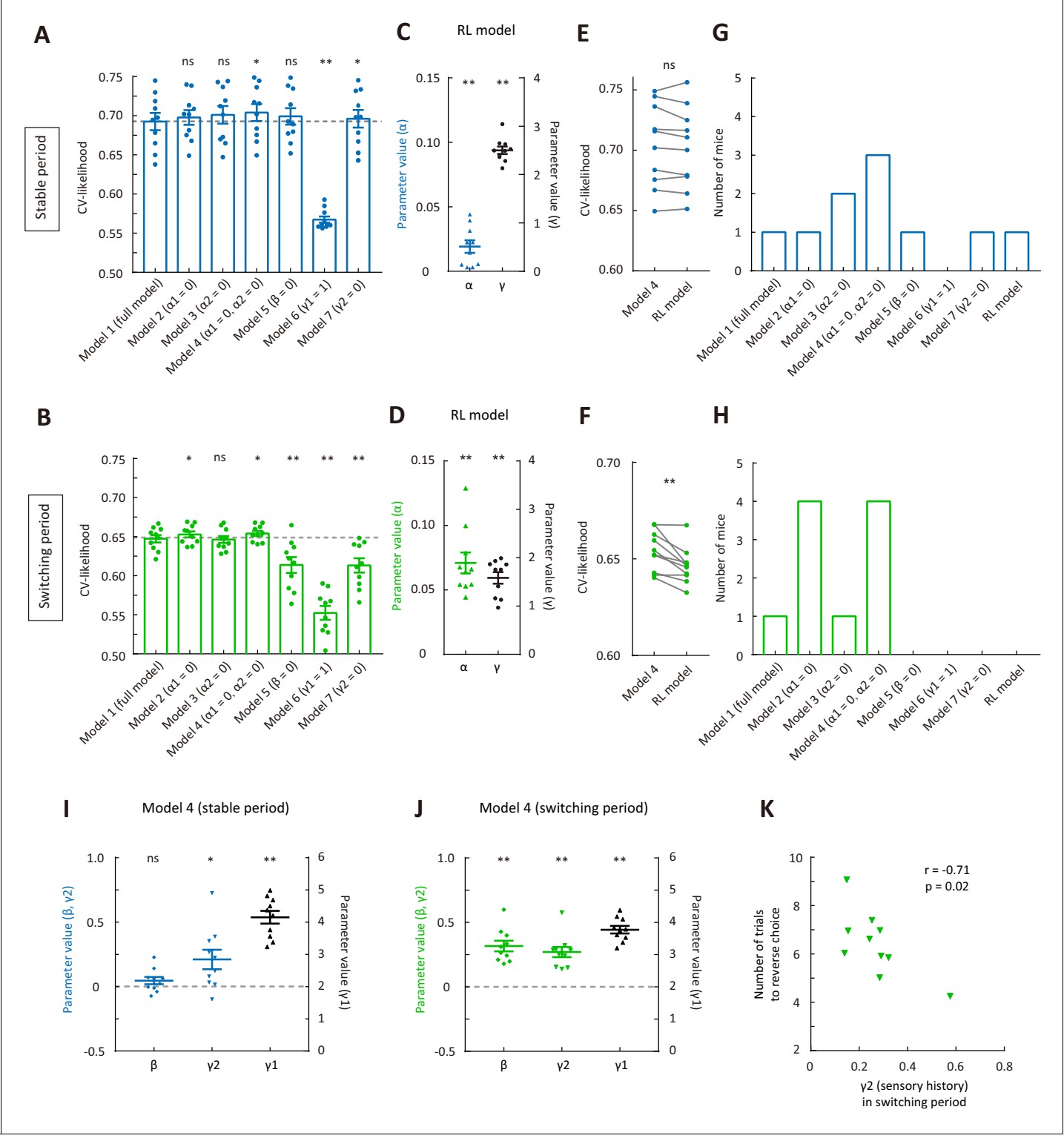

**Figure 3.** Comparison of cross-validated likelihood among different models. (**A**) CV likelihood of different variants of the dynamic-DC model for the stable period. For each of the reduced model variants (model 2 – model 7), we compared the CV likelihood with that of the full model. Dashed line, the CV likelihood averaged across 10 mice for the full model. (**B**) CV likelihood of different variants of the dynamic-DC model for the switching period. Similar to that described in (**A**). (**C**) Parameters of the RL model for the stable period. (**D**) Parameters of the RL model for the switching period. (**E**) Comparison of CV likelihood between the RL model and model 4 of the dynamic-DC model for the stable period. p=0.32, Wilcoxon signed rank test. (**F**) Comparison of CV likelihood between the RL model and model 4 of the dynamic-DC model for the switching period. p=$2 \times 10^{-3}$, Wilcoxon signed

*Figure 3 continued on next page*

*Figure 3 continued*

rank test. (**G**) Best model frequency (number of mice a model being the best model) for each model in the stable period. (**H**) Best model frequency for each model in the switching period. (**I**) Parameters of model four in the stable period. (**J**) Parameters of model four in the switching period. (**K**) Correlation between sensory history parameter $\gamma2$ (model 4) in the switching period and the number of trials to reverse choice. *p<0.05, **p<0.01, n = 10 mice, Wilcoxon signed rank test. Error bar,± SEM. For more details, see *Figure 3—figure supplements 1–7*. See *Figure 3—source data 1–3* for complete statistics.

The online version of this article includes the following source data and figure supplement(s) for figure 3:

**Source data 1.** Cross-validated likelihood for each model.
**Source data 2.** Parameter values for model 4.
**Source data 3.** Parameter values for the RL model.
**Source data 4.** Comparison of cross-validated likelihood among different models that were fitted to data in all trials.
**Source data 5.** Visualization of the simulated behavior for model 1 and model 4.
**Source data 6.** Parameter recovery analysis.
**Source data 7.** Right choice bias for reversing stimulus under two different conditions of sensory history.
**Figure supplement 1.** Comparison of cross-validated likelihood among different models that were fitted to data in all trials.
**Figure supplement 2.** Model simulation for model 1 and model four using the parameters of the model fitted to data in all trials.
**Figure supplement 3.** Simulated and actual performance for stimuli at SFs lower or higher than the SF of reversing stimulus.
**Figure supplement 4.** Parameter recovery for model 1 and model 4, using parameters of the model fitted to data in all trials.
**Figure supplement 5.** Parameter recovery for model 1 and model 4, using parameters of the model fitted separately to choices in switching and stable periods.
**Figure supplement 6.** Parameter recovery for model 1 and model 4, using a wide range of model parameters for the switching and stable periods.
**Figure supplement 7.** Comparison of right choice bias between two types of sensory history.

current stimulus) was most important for predicting choices during steady state of stimulus-action association. For the switching period, the CV likelihood for model 5 ($\beta = 0$), model 6 ($\gamma1 = 1$) and model 7 ($\gamma2 = 0$) was significantly reduced as compared to that of the full model (p=5.9 $\times$ 10$^{-3}$, 2 $\times$ 10$^{-3}$ and 5.9 $\times$ 10$^{-3}$, Wilcoxon signed rank test, *Figure 3B*), demonstrating that $\beta$, $\gamma1$ and $\gamma2$ were important parameters for explaining choices in the switching phase. For both the stable and switching periods, the CV likelihood for model 4 ($\alpha1 = 0$ and $\alpha2 = 0$) was significantly higher than that of the full model (p=0.037 and 0.02, Wilcoxon signed rank test, *Figure 3A and B*), suggesting that choice outcome was not useful to guide the mice's choice.

An alternative hypothesis is that the mouse might be updating the value functions of left and right choices separately, rather than comparing the current stimulus to one single decision boundary. We thus built a reinforcement learning (RL) model without an internal boundary. In the RL model, the expected value of left (right) choice is the learned value of left (right) choice weighted by current sensory stimulus, with the parameters $\alpha$ and $\gamma$ representing learning rate and stimulus weight, respectively. (Materials and methods, *Figure 3C and D*). We found that the CV likelihood of the RL model was significantly lower than that of the best dynamic-DC model (model 4: $\alpha1 = 0$ and $\alpha2 = 0$) in the switching period (p=2 $\times$ 10$^{-3}$, Wilcoxon signed rank test, *Figure 3F*), indicating that the behavioral strategy in the switching period was better accounted for by the model with a dynamic decision criterion.

For each mouse, we further determined which of the different types of model (the RL model and 7 variants of the dynamic-DC model) was the best model that yielded the largest CV likelihood. We found that model 4 of the dynamic-DC model ($\alpha1 = 0$ and $\alpha2 = 0$) was the best model for the greatest number of mice in both the stable and the switching periods (*Figure 3G and H*).

The above modeling analysis was based on fitting the models separately to choices in the stable and switching periods. We next used each of these models to fit choices in all trials (*Figure 3—figure supplement 1*). We found that the CV likelihood of model 1 or model four was both significantly higher than that of the RL model (*Figure 3—figure supplement 1*). Combining the result of model comparison and the histogram of the number of mice best fit by each model, we found that model 4 of the dynamic-DC model ($\alpha1 = 0$ and $\alpha2 = 0$) remained the winning model (*Figure 3—figure supplement 1*). We also visualized the simulated behavior for model 1 and model 4, using the parameters of the model fitted to data in all trials. As shown by *Figure 3—figure supplement 2*, both models could capture the dynamic change in performance for the reversing stimulus after the boundary switch. We found that the number of trials to reverse choice estimated from model one simulation tended to be larger than that estimated from the actual performance, whereas that

estimated from model four simulation matched well with the actual data (*Figure 3—figure supplement 2*). We also visualized the simulated psychometric curves. As shown by *Figure 3—figure supplement 2* for an example mouse, both model 1 and model four could capture the performance in the high-boundary block, but with an exaggerated right choice probability for SF at 0.044 or 0.065 cycles/° in the low-boundary block. For simulations using model parameters for the 10 mice in *Figure 1*, the simulated decision boundary in the high-boundary block was similar to the actual data (p=0.16 and 0.08 for model 1 and model 4, respectively, Wilcoxon signed rank test), whereas that in the low-boundary block was significantly lower than the actual data (p=0.002 for both model 1 and model 4, Wilcoxon signed rank test, *Figure 3—figure supplement 2*). This may be due to the fact that, in the model using a logistic function, the performance for a specific SF depended on the its distance from the boundary SF, so that the performance for low SFs was lower than that for high SFs in the low-boundary block and the performance for low SFs was higher than that for high SFs in the high-boundary block (*Figure 3—figure supplement 3*). For the mice's actual behavior, however, the performance difference between low and high SFs was more evident in the high-boundary block but less so in the low-boundary block (*Figure 3—figure supplement 3*). Thus, although the dynamic-DC model was not perfect to predict the psychometric curve, it could recapitulate the adaptive action selection for the reversing stimulus.

We further performed a parameter recovery analysis to check whether our fitting procedure can accurately estimate parameters. We simulated model 1 and model 4, respectively, using the parameters of the model fitted to data in all trials for the 10 mice in *Figure 1*. For both models, the recovered parameters matched the original parameters (*Figure 3—figure supplement 4*). We also simulated model 1 or model four using the parameters of the model fitted separately to choices in the stable and switching periods. This was performed using two sets of parameters: one was the model parameters for the 10 mice in *Figure 1*, another was a wider range of parameters (see Materials and methods). For both cases, there was good agreement between the recovered and original parameters (*Figure 3—figure supplement 5* and *Figure 3—figure supplement 6*).

The above analysis established that both model 1 and model 4 of the dynamic-DC model could recover original parameters from simulated data, and model four was the winning model to capture the adaptive behavior. As shown by *Figure 3I and J*, the sensory history parameter γ2 in model four was significantly larger than zero in both the stable and switching periods (p=0.014 and $2 \times 10^{-3}$, Wilcoxon signed rank test). When we computed a right choice bias for reversing stimulus under two different conditions of sensory history (previous low-frequency and previous high-frequency) (*Figure 3—figure supplement 7*), we found that, during the switching period, the right choice bias was significantly larger when the previous stimulus was at a lower than at a higher SF (p=$2 \times 10^{-3}$, Wilcoxon signed rank test), confirming the effect of sensory history as revealed by positive γ2. In addition, larger value of γ2 in the switching period (model 4) was associated with fewer number of trials to reverse choice (Pearson's r = −0.71, p=0.02, *Figure 3K*). Taken together, the above results suggest that the sensory history of non-reversing stimuli, rather than the choice-outcome history of reversing stimulus, was important for correct adaptation of decision boundary in this task.

## M2 inactivation impaired the reversing response after boundary switch

We next examined whether M2 activity is necessary for flexible action selection in the categorization task. We bilaterally injected AAV2/9-hSyn-hM4D-mCitrine in M2 to express the $G_i$-protein-coupled receptor hM4D (*Figure 4A*). Electrophysiological recordings validated that intraperitoneal injection of clozapine-N-oxide (CNO) could significantly reduce the firing rates of M2 neurons (p<0.001, n = 24 and 14 cells from two mice, respectively, Wilcoxon signed rank test, *Figure 4B*). To examine the effect of M2 inactivation, we compared the behavioral performance of mice between sessions with CNO and with saline injection (*Figure 4—figure supplement 1*), which were conducted in different days,~40 min before the behavioral task. To analyze the performance for the reversing stimulus before and after the boundary switch, we only included those sessions that contained ≥2 blocks. We found that chemogenetic inactivation of M2 significantly decreased the correct rate and response latency for the reversing stimulus during the switching period (p=0.022 and 0.019, two-way repeated-measures ANOVA followed by Sidak's multiple comparisons test), while did not significantly affect those during the stable period (p=0.87 and 0.47, n = 10, two-way repeated-measures ANOVA followed by Sidak's multiple comparisons test, *Figure 4C and D*). After the boundary switch, the number of trials to reverse choice for the reversing stimulus was greater in the CNO

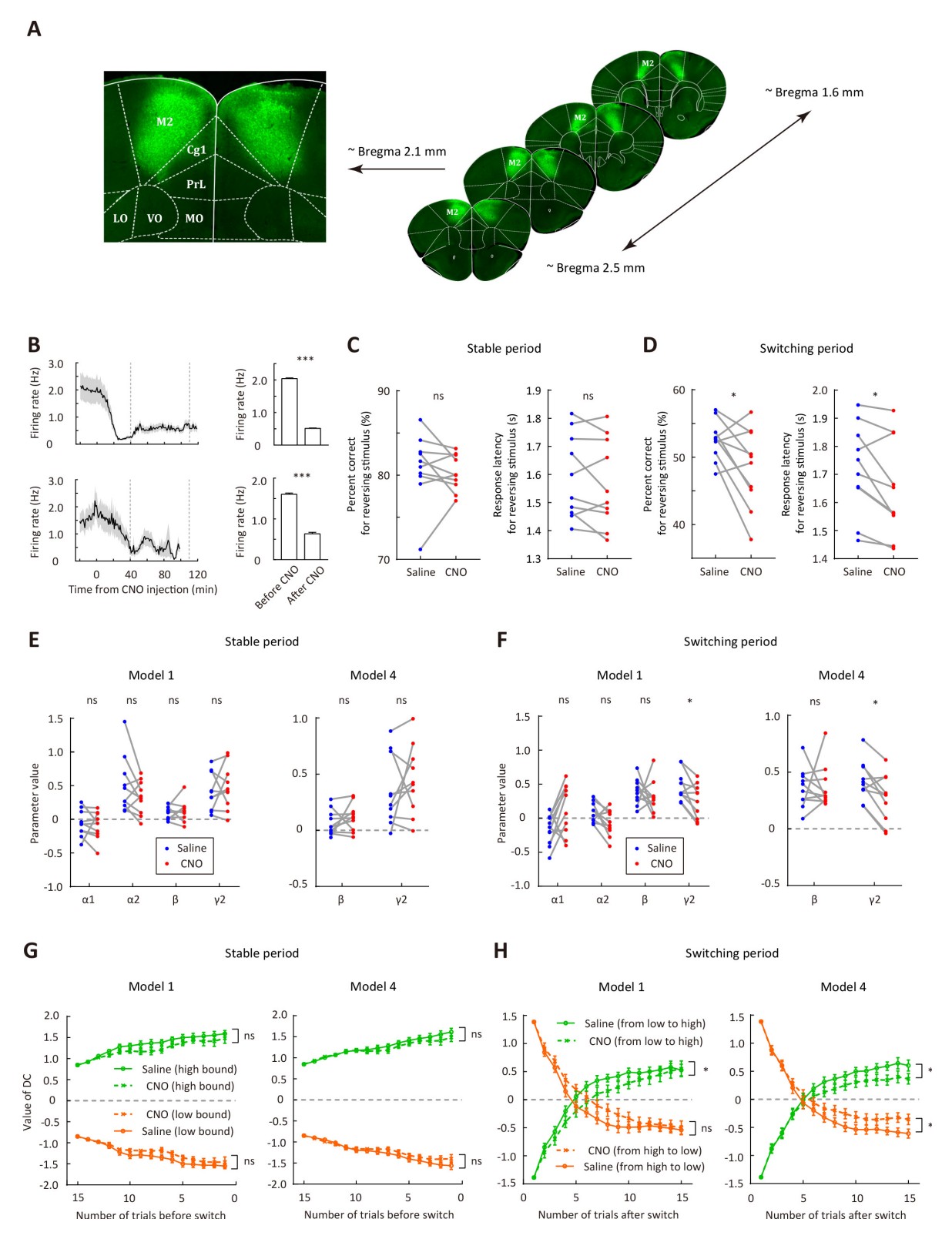

**Figure 4.** Bilateral inactivation of M2 impairs adaptive action selection in the switching period. (**A**) Representative fluorescence images showing the expression of AAV2/9-hSyn-hM4D-mCitrine in M2. (**B**) Firing rates of M2 neurons before and after CNO injection. Upper row, firing rates for 24 neurons recorded from one mouse (bar plot, mean firing rates 0–20 min before and 40–120 min after CNO injection). Lower row, firing rates for 14 neurons recorded from another mouse (bar plot, mean firing rates 0–20 min before and 40–100 min after CNO injection). The two vertical dashed lines indicate
*Figure 4 continued on next page*

*Figure 4 continued*

the time points corresponding to the start and end of a behavioral session, respectively. (C) Comparison of performance in the stable period between saline and CNO sessions. Left, correct rate for the reversing stimulus; right, response latency for the reversing stimulus. (D) Comparison of performance in the switching period between saline and CNO sessions, similar to that described in (C). (E) Parameters of the dynamic-DC model in the stable period. Left, model 1; right, model 4 ($\alpha 1 = 0$ and $\alpha 2 = 0$). (F) Parameters of the dynamic-DC model in the switching period. Left, model 1; right, model 4. (G) Comparison of DC curves in the stable period between saline and CNO sessions. Left, model 1; right, model 4. (H) Comparison of DC curves in the switching period between saline and CNO sessions. Left, model 1; right, model 4. *p<0.05, ***p<0.001. Wilcoxon signed rank test for (B). Two-way repeated measures ANOVA followed by Sidak's multiple comparisons test for (C and D). Two-way repeated measures ANOVA for (G and H). Data in (C−H) were from n = 10 mice. Shading and error bar,± SEM. For more details of the effect of chemogenetic manipulation, see *Figure 4—figure supplements 1–7*. See *Figure 4—source data 1–5* for complete statistics.

The online version of this article includes the following source data and figure supplement(s) for figure 4:

**Source data 1.** Effect of M2 inactivation on behavioral performance and model parameters.
**Source data 2.** Performance and model parameters for control mice with EGFP expressed in M2.
**Source data 3.** Parameter recovery analysis for saline and CNO sessions of M2 inactivation.
**Source data 4.** Effect of mPFC inactivation on behavioral performance and model parameters.
**Source data 5.** Effect of OFC inactivation on behavioral performance and model parameters.
**Figure supplement 1.** Effect of M2 inactivation on behavioral performance.
**Figure supplement 2.** Effect of CNO injection on the performance of EGFP-expressing control mice.
**Figure supplement 3.** M2 inactivation did not affect the performance for non-reversing stimuli (A), number of trials per block (B), number of switches per session (C) or the distance between internal decision boundaries (D).
**Figure supplement 4.** Effect of M2 inactivation on sensory history-dependent right choice bias.
**Figure supplement 5.** Parameter recovery for model 1 and model 4, using parameters of the model fitted separately to data in saline sessions and in CNO sessions.
**Figure supplement 6.** Effect of mPFC inactivation on behavioral performance.
**Figure supplement 7.** Effect of OFC inactivation on behavioral performance.

sessions (7.48 ± 1.19, mean ± SEM) than in the saline sessions (5.45 ± 0.40, mean ± SEM, p=0.049, n = 10, Wilcoxon signed rank test). For control mice with EGFP expressed in M2, data from saline and CNO sessions did not differ in the correct rate for reversing stimulus during either the stable or the switching period (*Figure 4—figure supplement 2*). On the other hand, M2 inactivation did not affect the correct rate for the non-reversing stimuli, the number of trials within a block, the number of switches per session or the distance between the two subjective categorical boundaries estimated from trials after the switching period (*Figure 4—figure supplement 3*), indicating that M2 inactivation did not cause general deficit in motor control or visual perception. Thus, bilateral M2 inactivation impaired adaptive action selection for the reversing stimulus specifically during the switching period when the mice needed to remap the stimulus-action association.

We next used the dynamic-DC model to fit the behavioral data of hM4D-expressing mice in saline sessions and CNO sessions, respectively, and compared the model parameters and DC values inferred from these parameters between saline and CNO sessions for both model 1 (full model) and model 4 ($\alpha 1 = 0$ and $\alpha 2 = 0$) (*Figure 4E−H*). We found that M2 inactivation did not cause significant changes in the model parameters (p>0.05, Wilcoxon signed rank test) or in the DC curves (p>0.05, two-way repeated measures ANOVA) during the stable period (*Figure 4E and G*), and did not significantly affect the value of $\gamma 1$ (weight of current stimulus) in either the stable or the switching period (p>0.05, Wilcoxon signed rank test). By contrast, M2 inactivation caused a significant decrease in $\gamma 2$ in both model 1 and model four during the switching period (p=0.02 and 0.014, Wilcoxon signed rank test, *Figure 4F*), suggesting an impairment in utilizing sensory history of the non-reversing stimuli to update DC. Analysis of choice bias during the switching period showed that M2 inactivation tended to reduce the choice bias difference between different conditions of sensory history (*Figure 4—figure supplement 4*), consistent with the decrease in $\gamma 2$. M2 inactivation also slowed down the change of DC during the switching period (*Figure 4H*). By simulating behavior with model 1 or model 4, we found that the recovered parameter $\gamma 2$ in the switching period showed a significant decrease in CNO sessions as compared to saline sessions (*Figure 4—figure supplement 5*), indicating that parameter change can be recovered from the model. For control mice with EGFP expressed

in M2, CNO injection did not cause a significant change in any of the model parameters for the stable or the switching period (*Figure 4—figure supplement 2*). Thus, a major effect of bilateral M2 inactivation was an impairment in updating internal decision criterion according to the sensory history of non-reversing stimuli during the switching period.

Prefrontal cortical regions, including the medial prefrontal cortex (mPFC) and the orbitofrontal cortex (OFC), have been implicated in flexible behavior (*Birrell and Brown, 2000*; *Dias et al., 1996*; *Duan et al., 2015*; *Floresco et al., 2006*; *Izquierdo et al., 2017*; *Ragozzino, 2007*; *Ragozzino et al., 1999*; *Stefani et al., 2003*). We thus also examined the role of prefrontal cortex in the flexible visual categorization task. We found that bilateral inactivation of mPFC or OFC did not cause significant change in the performance for the reversing stimulus, the distance between internal categorical boundaries or number of switches per session (p>0.05, Wilcoxon signed rank test, mPFC: n = 13 mice, OFC: n = 9 mice, *Figure 4—figure supplement 6* and *Figure 4—figure supplement 7*), although OFC inactivation tended to increase the number of trials per block (*Figure 4—figure supplement 7*). Inactivation of mPFC or OFC did not affect the parameters in the dynamic-DC model (*Figure 4—figure supplement 6* and *Figure 4—figure supplement 7*). Thus, the impaired performance for the reversing stimulus during the switching period was specific to M2 inactivation.

## Information about choice and sensory history encoded by M2 neurons

We next examined the responses of M2 neurons in behaving mice (see Materials and methods, *Figure 5—figure supplement 1*). Mice used for electrophysiological recordings were also found to adopt behavioral strategies (*Figure 5—figure supplement 2*) similar to those mice in *Figure 2* and *Figure 3*. For each M2 neuron, we analyzed the firing rates around the holding period, during which the mice held their heads in the central port viewing the stimulus and presumably forming a decision.

As M2 inactivation-induced impairment in performance for the reversing stimulus during the switching period was largely due to a reduced influence of sensory history (*Figure 4*), we examined the choice- and sensory history-related responses of M2 neurons during the reversing-stimulus trials (*Figure 5A*−D). To compare neural responses between different choices (left vs right) or between different sensory history (previous low-frequency vs previous high-frequency), we used the receiver operating characteristic (ROC) analysis (*Green and Swets, 1966*) to measure auROC (area under the ROC curve) (*Figure 5E*−J), which was used to quantify ROC preference. As the right choice bias of the mice was significantly larger when the previous stimulus was at a lower than at a higher frequency (*Figure 5—figure supplement 3*), to understand the relationship between preference for choice and preference for previous stimulus, we used the responses of left and right choices to compute choice preference. We computed the ROC preference as $2 \times (\text{auROC} - 0.5)$, which ranged from −1 to 1 (*Feierstein et al., 2006*). For left-right choice preference, negative (positive) values indicate that the firing rates in current trial were higher when the choice is left (right). To compute the preference for previous stimulus, we divided the responses of current reversing-stimulus trials into two groups, in which the previous frequency was higher and lower than the frequency of the reversing stimulus, respectively. For previous-stimulus preference, negative (positive) values indicate that the firing rates in current trial were higher when the previous non-reversing stimulus was at a low (high) SF.

Because neither the choice preference nor the previous-stimulus preference was different between neurons recorded from the left and right hemispheres (*Figure 5—figure supplement 4*), we combined neurons recorded in both hemispheres. We found that the choice preferences in correct and wrong trials were positively correlated (stable period: Pearson's r = 0.36, p=$1.0 \times 10^{-4}$; switching period: Pearson's r = 0.24, p=$9.3 \times 10^{-3}$; n = 113, *Figure 5K*), indicating that M2 activity during stimulus-viewing period reflects the orienting choice of the mice, consistent with the previous report (*Erlich et al., 2011*). We next examined the relationship between choice preference and previous-stimulus preference. For correct trials, the preference for left-right choice in current trial showed a significant negative correlation with the preference for stimulus in the last trial (stable period: Pearson's r = −0.87, p=$3.2 \times 10^{-23}$; switching period: Pearson's r = −0.92, p=$9.5 \times 10^{-31}$; n = 72, *Figure 5L*), and such effect could be observed for M2 neurons recorded in both hemispheres (*Figure 5—figure supplement 5*). Using a sliding window of 50 ms, we found that the negative correlation was observed in all time bins, including those time bins between central port entry and stimulus onset. For the sliding window analysis of correlation between the preference for left-right

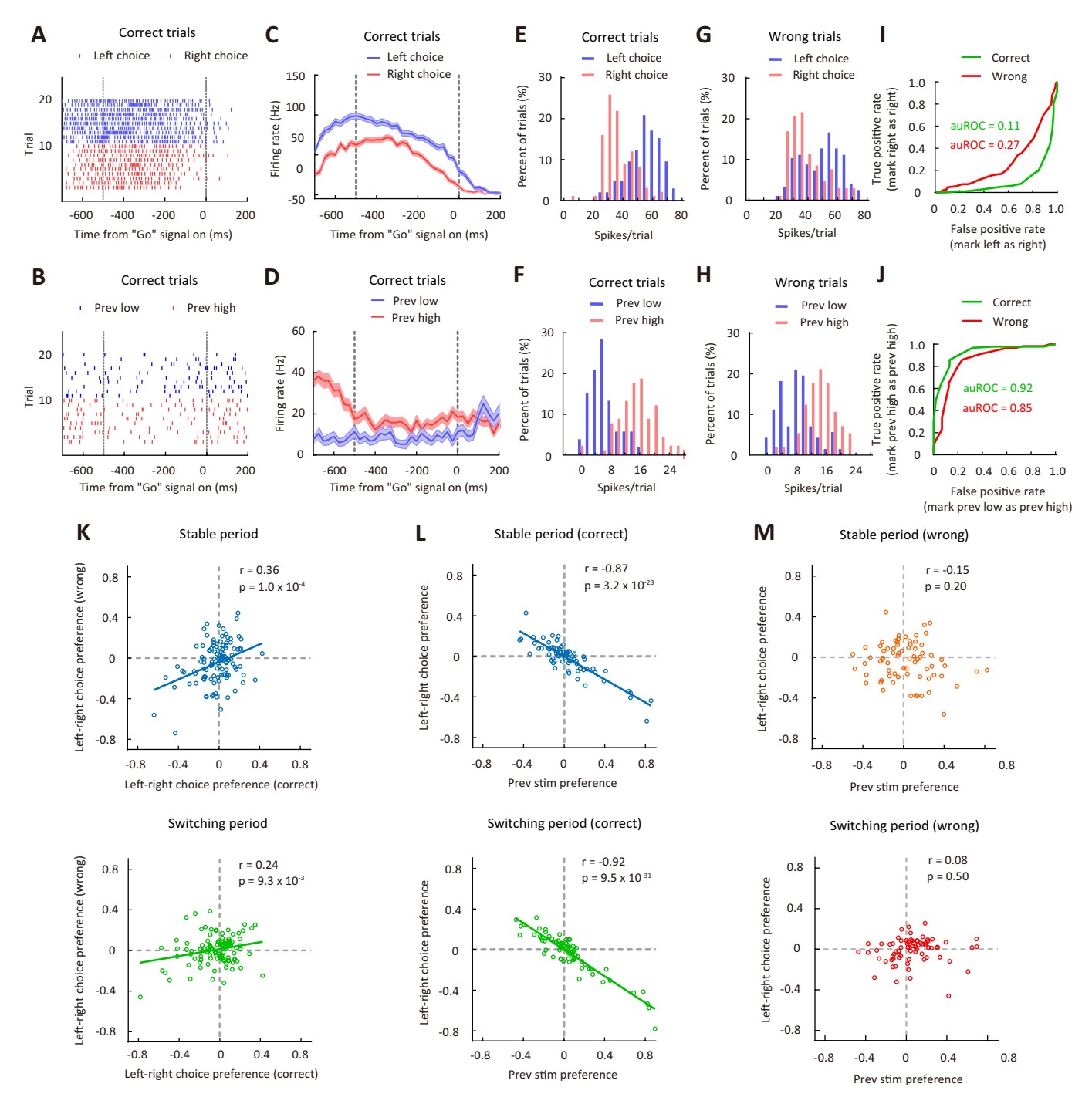

**Figure 5.** Correlation between choice preference and previous-stimulus preference for M2 neurons. (**A**) Spike raters of an example M2 neuron in 20 reversing-stimulus trials, grouped by left or right choice. The two vertical dashed lines indicate the time of stimulus onset and 'Go' signal onset, respectively. (**B**) Spike raters of another example M2 neuron in 20 reversing-stimulus trials, grouped by previous stimulus (previous low or previous high SF). (**C**) PSTHs of the M2 neuron in (**A**) in correct reversing-stimulus trials, grouped by left or right choice. (**D**) PSTHs of the M2 neuron in (**B**) in correct reversing-stimulus trials, grouped by previous stimulus (previous low or previous high SF). (**E**) Frequency histogram of left-choice and right-choice responses from the M2 neuron in (**A**) in correct trials. (**F**) Frequency histogram of previous-low-frequency and previous-high-frequency responses from the M2 neuron in (**B**) in correct trials. (**G**) Frequency histogram of left-choice and right-choice responses from the M2 neuron in (**A**) in wrong trials. (**H**) Frequency histogram of previous-low-frequency and previous-high-frequency responses from the M2 neuron in (**B**) in wrong trials. (**I**) ROC curves for the two pairs of response distributions illustrated in (**E**) and (**G**), respectively. (**J**) ROC curves for the two pairs of response distributions illustrated in (**F**) and

*Figure 5 continued on next page*

*Figure 5 continued*

(H), respectively. (K) Left-right choice preference in correct and wrong trials was significantly correlated. n = 113 neurons. Upper, stable period; lower, switching period. (L) Left-right choice preference and previous-stimulus preference were significantly correlated in correct trials. n = 72 neurons. Upper, stable period; lower, switching period. (M) Left-right choice preference and previous-stimulus preference were not significantly correlated in wrong trials. n = 72 neurons. Upper, stable period; lower, switching period. r, Pearson's correlation coefficient. Shading in (C and D), ± SEM. See *Figure 5—figure supplements 1–7* for more details of the recording and data analysis. See *Figure 5—source data 1* for complete statistics.

The online version of this article includes the following source data and figure supplement(s) for figure 5:

**Source data 1.** Choice preference and previous-stimulus preference for M2 neurons.
**Source data 2.** Model parameters and right choice bias for the mice used for electrophysiologcial recordings.
**Source data 3.** Choice preference and previous-stimulus preference for M2 neurons recorded from the left and right hemispheres, respectively.
**Source data 4.** Sliding window analysis of the correlation between choice preference and previous-stimulus preference of M2 neurons in correct trials.
**Figure supplement 1.** Electrophysiological recordings from M2.
**Figure supplement 2.** Model fitting for the behavioral data of mice used for electrophysiological recordings.
**Figure supplement 3.** Comparison of right choice bias between two types of sensory history for mice used for electrophysiological recordings.
**Figure supplement 4.** Choice preference and previous-stimulus preference for M2 neurons recorded from the left and right hemispheres.
**Figure supplement 5.** The correlation between choice preference and previous-stimulus preference in correct trials for M2 neurons in left and right hemispheres, respectively.
**Figure supplement 6.** Sliding window analysis of the correlation between choice preference and previous-stimulus preference for correct trials.
**Figure supplement 7.** Choice preference (ipsilateral vs contralateral) was not significantly correlated with preference for previous stimulus (low-frequency vs high-frequency) for M2 neurons.

choice in current trial and the preference for stimulus in the trial before last (*Figure 5—figure supplement 6*), the statistical significance of the correlation diminished in most time bins. Thus, the result suggests that the choice preference of M2 neurons is strongly influenced by stimulus history, and the preference for choice in current trial is coupled to the preference for stimulus in the last but not earlier trial. For wrong trials, however, the correlation between choice preference and previous-stimulus preference was not significant (stable period: Pearson's r = −0.15, p=0.2; switching period: Pearson's r = 0.08, p=0.5; n = 72, *Figure 5M*). Furthermore, the preference for ipsilateral-contralateral choice in current trial did not exhibit a significant correlation with the preference for stimulus in last trial, for either correct or wrong trials (*Figure 5—figure supplement 7*). Thus, the left-right choice preference of M2 neurons was intimately related to the preference for previous stimulus in correct trials, consistent with the finding that sensory history of previous non-reversing stimuli was important for correct action selection in the task.

We further trained linear support vector machine (SVM) classifier to decode upcoming choice or sensory history from M2 responses to the reversing stimulus (see Materials and methods). The analysis was applied separately to correct (or wrong) trials in the stable and switching periods. For each type of choice (sensory history) under each condition, we randomly sampled 10 trials from each neuron to form a resampled dataset, trained the classifier on 75% of the resampled data, and tested it on the remaining 25% of the resampled data. This cross-validation procedure was repeated 100 times and an averaged prediction accuracy was computed. The resampling procedure to compute prediction accuracy was repeated for 1500 times. As shown by *Figure 6A*, the accuracy of predicting choice was significantly higher in the switching than in the stable period for correct trials (p<1.0 $\times$ 10$^{-4}$, Wilcoxon rank sum test). For both the stable and switching periods, the accuracy of predicting choice was higher for correct trials than for wrong trials (p<1.0 $\times$ 10$^{-4}$, Wilcoxon rank sum test, *Figure 6A*). When we examined the accuracy for classifying the previous stimulus as a lower or a higher frequency, we also found that the accuracy was significantly higher in the switching than in the stable period for correct trials (p<1.0 $\times$ 10$^{-4}$, Wilcoxon rank sum test), and the accuracy for the switching period was higher in correct than in wrong trials (p<1.0 $\times$ 10$^{-4}$, Wilcoxon rank sum test, *Figure 6B*). Thus, the results demonstrate that representations of upcoming choice and sensory history in M2 were stronger during the switching period, when mice were faced with a demand to switch stimulus-action association, than during the stable period. This corresponds with the causal contribution of M2 in controlling action selection when the stimulus-action association was remapped.

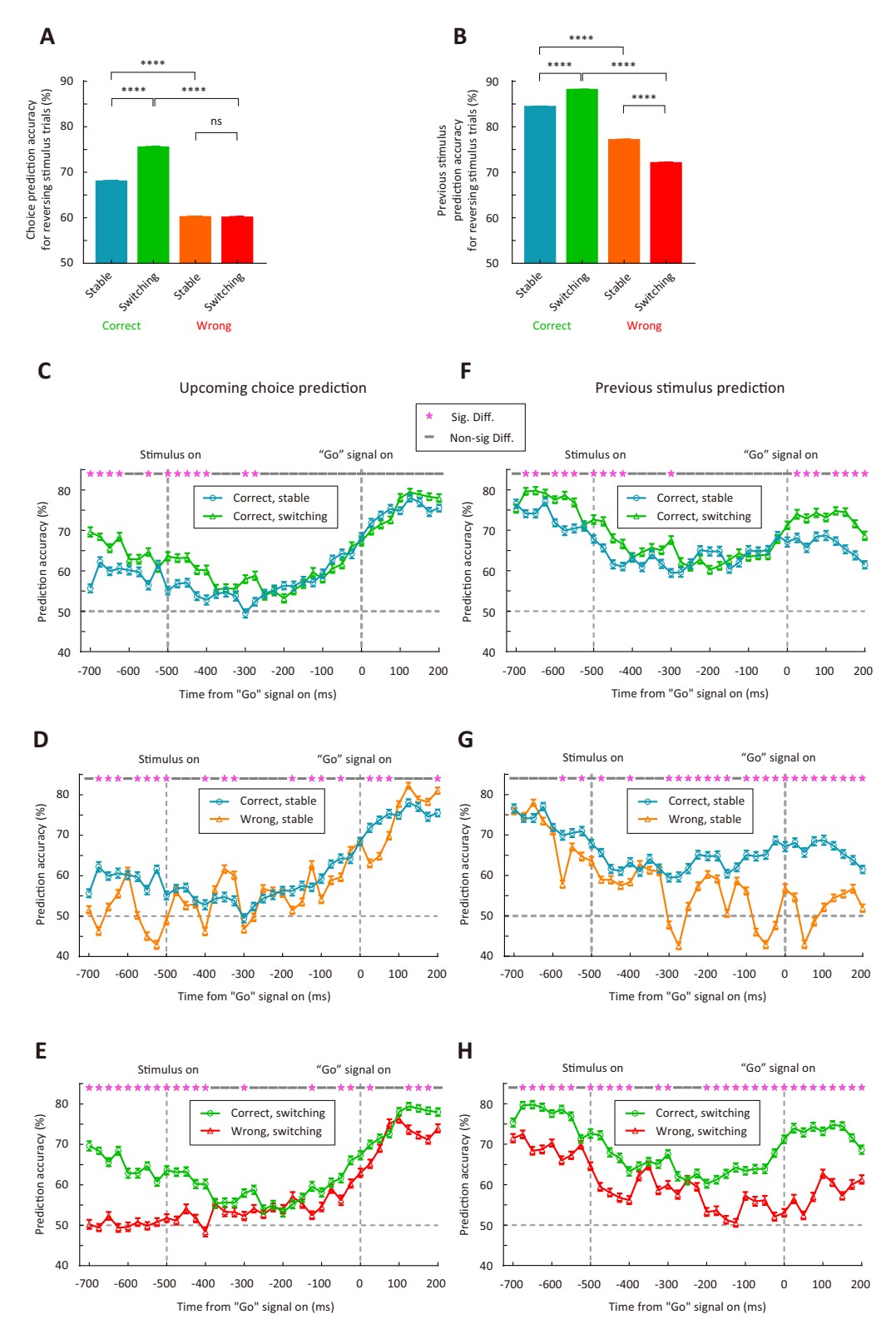

**Figure 6.** M2 neurons encode upcoming choice and previous stimulus more accurately in the switching than in the stable period. (A) Prediction accuracy for upcoming choice in response to the reversing stimulus. n = 1500 resampled datasets, 113 neurons. (B) Prediction accuracy for stimulus in previous trial. n = 1500 resampled datasets, 72 neurons. (C–H) Temporal dynamics of decoding accuracy. Time 0 is the time of 'Go' signal onset, and −700 ms is the time of central port entry. (C−E) Prediction accuracy for upcoming choice in response to the reversing stimulus. (F−H) Prediction

*Figure 6 continued on next page*

*Figure 6 continued*

accuracy for stimulus in previous trial. For (A and B), ****, $p<1.0 \times 10^{-4}$, Wilcoxon rank sum test. For (C–H), * marks the time bins with $p<0.05$, two-way ANOVA followed by Sidak's multiple comparisons test. See *Figure 6—source data 1* for complete statistics.

The online version of this article includes the following source data for figure 6:

**Source data 1.** SVM analysis of M2 activity.

To quantify the dynamics of prediction accuracy, we next trained and tested SVM classifier on each 50 ms time bin with a 25 ms moving window (*Figure 6C*−H). For decoding upcoming choice, we found that the prediction accuracy tended to increase over time during the holding period (*Figure 6C*−E). For decoding upcoming choice in correct trials, higher accuracy in the switching than in the stable period was mostly observed within a ± 200 ms window around stimulus onset (*Figure 6C*). For decoding previous stimulus, the prediction accuracy was highest around the time of central port entry (*Figure 6F*−H), consistent with the representation of trial history. For decoding previous stimulus in correct trials, higher accuracy in the switching than in the stable period was evident both around the time of central port entry and after the onset of 'Go' signal (*Figure 6F*). Although the prediction accuracy for previous stimulus was decreased after stimulus onset, the decrease was more evident in wrong trials, resulting in higher accuracy in correct than in wrong trials (*Figure 6G and H*). Thus, the representations of upcoming choice and previous stimulus by M2 neurons exhibited different dynamics, and temporal integration of both information may be important for the action selection of mice.

## Discussion

Using behavioral modeling to analyze choices in a flexible visual categorization task, we found that mice depended on sensory history to correctly change stimulus-action association when categorical boundary switched between blocks of trials. Chemogenetic manipulation showed that M2 activity was specifically required for correct choices during remapping of stimulus-action association, but not when the sensorimotor association was stable. We further found that the representations of upcoming choice and sensory history by M2 neurons were stronger when the sensorimotor association needed to be flexibly adjusted. Thus, the choice- and sensory history-related signals in M2 are adaptive to task requirement, which may account for the important role of M2 in adaptive choice behavior during flexible stimulus categorization.

Understanding behavioral strategy using computational modeling is important for studying the neural basis of behavior (*Churchland and Kiani, 2016*; *Krakauer et al., 2017*). In our task of visual categorization, the mice were required to switch choice to the reversing stimulus several times within a single session. We thus used a probabilistic choice model in which the decision variable was determined by the comparison between current sensory input and a dynamic decision criterion. Previous studies showed that trial history impaired performance in typical perceptual decision tasks (*Abrahamyan et al., 2016*; *Jiang et al., 2019*). For the task in current study, however, because the probability of the non-reversing stimulus at the lowest (or highest) frequency differed between blocks, sensory history could be used as a prominent cue indicating a change in categorization contingency. Our behavioral model and choice bias analysis revealed that sensory history significantly contributed to adaptive choices during the switching period. Although a change in the outcome of response to the reversing stimulus may also be used for guiding adaptive choices, the model analysis showed that action outcome history had a negligible effect on updating DC in the switching period, particularly for those well-trained mice that had been trained for 3–4 months before data collection (*Figure 2*). Unlike the mice in *Figure 2*, those mice in *Figure 4* were trained with a shorter period of time (~40 d) before we measured the effect of chemogenetic manipulation and those in *Figure 5—figure supplement 2* were measured with a narrower range of SFs. The behaviors of mice in *Figure 4* (saline sessions) and *Figure 5—figure supplement 2* exhibited a win-shift strategy ($\alpha 1 < 0$) in the switching period, suggesting that the effect of choice-outcome history may depend on length of training or stimulus difficulty. We also showed that, for the flexible visual categorization task in this study, the model with a dynamic DC could better account for the adaptive choices than a RL model in which sensory stimuli were used in the computation of expected values of left and right choices. Future work may further compare the dynamic DC model with other forms of RL model. In our study,

the change in stimulus statistics associated with the boundary switch likely promoted a strategy that involved sensory history during adaptive action selection. On the other hand, it is possible that the influence of outcome may occur at a slower timescale, which may not be well captured by our model based on the trial-by-trial analysis. In the future, applying computational models that consider the influence of sensory history and outcome at different timescales may allow us to better understand the process of adaptive action selection.

Categorical decision involves action selection that recruits the premotor cortex (*Ashby and Maddox, 2005*). In primates, the pre-supplementary motor area is involved in the switch of actions (*Isoda and Hikosaka, 2007*; *Rushworth et al., 2002*). Rodent M2, which is a putative homolog of primate premotor cortex, supplementary motor regions and frontal eye field, has been shown to play a critical role in action selection guided by sensory stimuli, memory or prior actions (*Barthas and Kwan, 2017*; *Erlich et al., 2011*; *Gilad et al., 2018*; *Goard et al., 2016*; *Guo et al., 2014*; *Itokazu et al., 2018*; *Li et al., 2015*; *Makino et al., 2017*; *Murakami et al., 2017*; *Ostlund et al., 2009*; *Siniscalchi et al., 2016*). M2 neurons have been found to encode information about trial history as well as upcoming choice (*Hattori et al., 2019*; *Jiang et al., 2019*; *Scott et al., 2017*; *Siniscalchi et al., 2019*; *Sul et al., 2011*; *Yuan et al., 2015*). In our study, we found that the left-right choice preference of M2 neurons was well correlated with the preference for previous stimulus in correct trials, suggesting that M2 neurons could integrate sensory history to generate choice signal. Previous studies also showed that M2 activity is task dependent. For instance, M2 responses are modulated by effector, task context and action outcome (*Erlich et al., 2011*; *Kargo et al., 2007*; *Murakami et al., 2014*; *Siniscalchi et al., 2019*). A recent study found that M2 activity patterns differ between conditions of cue-guided and nonconditional actions, and bilateral muscimol inactivation of M2 slowed the transition from action-guided to sound-guided response, without affecting the high performance of sound-guided trials after the transition (*Siniscalchi et al., 2016*). Consistent with this study, we found that bilateral inactivation of M2 slowed the adaptive adjustment of action selection following the boundary switch, without affecting choice behavior in steady state. The effect of M2 inactivation on adaptive activation selection is also consistent with our finding that the representations of choice and history information by M2 neurons were stronger during the switching period.

Mouse M2 occupies a wide range of cortical area, spanning from AP 2.5 mm to AP −1 mm along the rostral-caudal axis (*Barthas and Kwan, 2017*; *Franklin and Paxinos, 2007*). In our study, the effect of bilateral inactivation of M2 (AP 2.0 mm, ML 0.75 mm) on adaptive action selection was modest, which may be partly due to the fact that the virus infected only a limited part of M2. Alternatively, it is possible that other parts of M2 or other regions also contribute to the adaptive action selection. We found that inactivation of M2 but not mPFC or OFC affected the performance for reversing stimulus in the switching period, suggesting that sensory-history dependent adaptive action selection may specifically require M2 activity. Of course, it should be noted that our chemogenetic manipulation did not produce a complete inactivation of neuronal activity and thus the negative effect of mPFC/OFC inactivation may need to be further confirmed using experiments with more complete inactivation. Another limitation of our chemogenetic manipulation is the lack of precise temporal control of M2 inactivation. Future studies using optogenetics may allow to examine the temporal specificity of the effect of M2 inactivation. It is also of interest to further examine the circuit mechanism for adaptive encoding of choice signal in M2.

## Materials and methods

### Key resources table

| Reagent type (species) or resource | Designation | Source or reference | Identifiers | Additional information |
|---|---|---|---|---|
| Strain, strain background *Mus musculus* | C67BL/6 | Slac Laboratory Animal | N/A | |
| Chemical compound, drug | clozapine-N-Oxide | Sigma-Aldrich Corporation | C0832-5MG | |

*Continued on next page*

*Continued*

| Reagent type (species) or resource | Designation | Source or reference | Identifiers | Additional information |
|---|---|---|---|---|
| Software, algorithm | Offline Sorter | Plexon | https://plexon.com | |
| Software, algorithm | Prism | GraphPad | https://www.graphpad.com/scientificsoftware/prism/; RRID:SCR_00279 | |
| Software, algorithm | MATLAB | Mathworks | https://www. mathworks.com/; RRID:SCR_001622 | |

## Animals

Animal use procedures were approved by the Animal Care and Use Committee at the Institute of Neuroscience, Chinese Academy of Sciences (approval number NA-013–2019), and were in accordance with the guidelines of the Animal Advisory Committee at the Shanghai Institutes for Biological Sciences. Data were from a total of 68 male adult C57BL/6 mice (3–12 months old), in which 11 were used for electrophysiological recordings during behavioral task. The mice were housed in groups of four to six per cage (mice for chronic extracellular recordings were housed individually). The mice were deprived of water in the home cage and obtained water reward during daily behavior sessions. On days that mice did not perform the task, restricted water access (~1 ml) was provided each day. The mice were kept on a 12 hr light/12 hr dark cycle (lights on at 7:00 am). All sessions were performed in the light phase.

## Behavior and visual stimuli

All training and experiments took place in a custom-design behavioral chamber with three ports (*Long et al., 2015*), controlled by custom Matlab (Mathworks, Natick, USA) scripts and digital I/O devices (PCI-6503, National Instruments Corporation, Austin, USA). Visual stimuli were presented on a 17' LCD monitor (Dell E1714S, max luminance 100 cd/m$^2$) placed 10 cm away from the front wall of the chamber. Gamma correction was used to calibrate the monitor to establish a linear relationship between program-controlled pixel intensity and actual luminance. The monitor subtended 126° × 113° of visual space, assuming that the mouse's head was at the central port facing the stimulus. A yellow light-emitting diode (LED) was placed at the central port to provide the 'Go' signal.

The mice initiated a trial by nose-poking into a hole in the central port. A full-field visual stimulus was presented on the monitor after 200 ms of trial initiation, and the mice were required to continue staying in the central port for at least 500 ms until the yellow LED ('Go' signal) lighted up. The mice were required to compare the current stimulus to an internal decision boundary and reported their choice by going to one of the two side ports. The rewarded side port was on the left (right) if the stimulus SF was lower (higher) than a categorical boundary. For each trial, the visual stimulus was presented until the mouse chose one of the two side ports, and the 'Go' signal was turned off at the same time as stimulus offset. Gray screen of mean luminance was used between trials.

The mice were trained to perform flexible visual categorization task using the following steps. In step 1 (2 d), the central port was blocked, and mice could collect water reward by nose-poking to the left and right ports alternatively. In step 2 (2 d), one of the side ports was blocked alternatively between daily sessions. The mice learned to initiate a trial by poking its nose into the central port, then go to the unblocked side port to receive water reward. In step 3 (3–10 d), the task was similar to that in step two except that the mouse was required to hold its head in the central port. The required holding time was at 50 ms initially and increased by 75 ms each time the mouse succeeded holding in 75% of the previous 20 trials. The training continued until the successful holding time reached 1000 ms in 75% of trials in a session. In step 4 (4–10 d), all ports were open, and the mouse initiated a trial by nose-poking into the central port. After the mouse held its head for 200 ms, a full-field static grating (vertical orientation) at a low SF (0.03 cycles/°) or a high SF (0.3 cycles/°) was presented until the mouse entered one of the two side ports. If the mouse chose the left (right) side port for a low (high) frequency, it was rewarded by 3–4 µl water. Thus, in this step the mouse learned to discriminate between a low and a high SF. In step 5 (2–15 d), each daily session consisted of two types of blocks. Stimuli in the low-boundary block consisted of gratings at 0.03 and 0.095 cycles/°, and those in the high-boundary block were at 0.095 and 0.3 cycles/°. Each block consisted of at least

60 trials. For both types of blocks, a low (high) frequency stimulus indicated that choosing the left (right) port was considered as correct and would be rewarded. After the performance for the reversing stimulus (0.095 cycles/°) reached 70% or 80% in the last 10 reversing-stimulus trials, the boundary switched and the mouse was required to reverse its choice for the reversing stimulus. Thus, in this step the mouse learned to switch stimulus-action association for the reversing stimulus at 0.095 cycles/°. In step 6 (3–7 d for most mice), the task was similar to that in step five except that visual stimuli consisted of 7 SFs (0.03, 0.044, 0.065, 0.095, 0.139, 0.204, and 0.3 cycles/°) in both types of blocks. For the low-boundary block, gratings at 0.03 and 0.095 cycles/° were presented for 90% of trials, gratings at the other frequencies were presented for 10% of trials, and the boundary frequency was between 0.065 and 0.095 cycles/°. For the high-boundary block, gratings at 0.095 and 0.3 cycles/° were presented for 90% of trials, and the boundary frequency was between 0.095 and 0.139 cycles/°. Thus, the stimulus statistics differed between blocks. After step 6, the mouse was used for data collection of behavioral experiment. For behavioral data of mice in *Figure 1 – Figure 4*, the visual stimuli were the same as those in step 6.

For behavioral experiments in control mice with EGFP expressed in M2 (*Figure 4—figure supplement 2*), in mice with chemogenetic inactivation of mPFC or OFC (*Figure 4—figure supplement 6* and *Figure 4—figure supplement 7*) and in mice used for electrophysiological recordings (*Figure 5* and *Figure 6*), the SFs of the gratings were at 0.06, 0.073, 0.09, 0.11, 0.134, 0.164, and 0.2 cycles/°. In the low-boundary block, gratings at 0.06 and 0.11 cycles/° were presented for 90% of trials, and the boundary frequency was between 0.09 and 0.11 cycles/°. In the high-boundary block, gratings at 0.11 and 0.2 cycles/° were presented for 90% of trials, and the boundary frequency was between 0.11 and 0.134 cycles/°.

For mice not used for electrophysiological recordings, the behavior of each mouse was measured for 9.41 ± 1.42 (mean ± SD) sessions. For mice used for electrophysiological recordings, the behavior of each mouse was measured for 23 ± 5.2 (mean ± SD) sessions.

## Surgery

For chemogenetic inactivation experiments, mice were injected with virus before behavioral training. Mice were anesthetized with a cocktail of midazolam (5 mg/kg), medetomidine (0.5 mg/kg) and fentanyl (0.05 mg/kg) injected intraperitoneally before surgery, then head-fixed in a stereotaxic apparatus. Their body temperature was maintained at 37°C. Eye drops and eye ointment were applied to the eyes to prevent from drying. The incision site was treated with lidocaine jelly. Two craniotomies (~ 0.8 mm diameter) were performed bilaterally above M2 (AP 2.0 mm, ML ±0.75 mm), mPFC (AP 2.3 mm, ML ±0.3 mm) or OFC (AP 2.6 mm, ML ±1.0 mm). For virus injection, the bones were not cracked, only thinned enough to allow easy penetration by a borosilicate glass pipette with a tip diameter of ~40–50 μm. A total of 300 nl AAV2/9-hSyn-hM4D-mCitrine (or AAV2/8-CamKIIa-EGFP-3Flag-WPRE-SV40pA for control mice, which were randomly assigned among cagemates) was injected at a depth of 900 μm for M2 (at 1350 μm for mPFC and at 2100 μm for OFC) using a syringe pump (Pump 11 Elite, Harvard Apparatus, Holliston, USA). After the virus injection, the pipette was left in place for 10–15 min before retraction. The mice were given carprofen (5 mg/kg) subcutaneously after the surgery.

Chronic electrodes were implanted after the mice were fully trained. At least 2 days before the surgery, the mice were granted free access to water. The mice were anesthetized and prepared using the same procedures described above. A grounding screw was implanted at a site posterior to Lamda. A craniotomy was made above the left or right M2 (AP 2.0 mm, ML 0.75 mm), and a portion of the dura was removed. The cortical surface was applied with artificial cerebrospinal fluid. A 16-sites silicon probe (A1 ×16–3 mm-50-177-CM16LP, NeuroNexus Technologies, Ann Arbor, USA) was lowered into the brain to a depth of 900 μm with a micromanipulator (Siskiyou Corporation, Grants Pass, USA). The craniotomy was covered with a thin layer of silicone elastomer Kiwi-Cast, followed by another layer of silicone elastomer Kiwi-Sil (World Precision Instruments, Sarasota, USA). A head-plate was placed on the skull to facilitate later handling of the animal (attaching the headstage to the probe connector before each recording session), and cyanoacrylate tissue adhesive (Vetbond, 3M, Saint Paul, USA) was applied to the skull surface to provide stronger fixation. After the tissue adhesive cured, several layers of dental acrylic cement were applied to fix the whole implant in place. A ferrule of optical fiber was partially embedded in the dental cement (above the grounding screw) to provide guidance to a rotary joint (Doric Lenses, Quebec, Canada). Mice were allowed to

recover from the surgery for at least 7 days before water-restriction and behavioral training. For mice implanted with electrodes, they were returned to step 4 of the training to get used to moving around with the implants and headstage cable before the recordings started.

## Chemogenetic inactivation

We used the system of designer receptor exclusively activated by designer drug (DREADD) (*Stachniak et al., 2014*) to inactivate M2, mPFC or OFC. At ~40 min before each daily session, the mice were briefly anesthetized with isoflurane (4%) and received an intraperitoneal injection of clozapine-N-Oxide (CNO) (1.5 mg/kg) or saline. The concentration of CNO solution was adjusted such that each mouse received ~100 µl of solution from the injection. The mice were allowed to recover individually in a cage before behavioral testing. We verified in anesthetized mice that the effect of DREADD inactivation on M2 neuronal spiking lasted for 2 hr. Because some SFs were presented for 10% of trials and the number of switches per session was limited (<7 for most mice), we measured 8.7 ± 0.95 (mean ± SD) sessions for saline injection (or CNO injection), so that we could obtain enough number of trials for psychometric curve analysis and modelling analysis.

## Electrophysiological recording

Neural signals were amplified and filtered using the Cerebus 32-channel system (Blackrock Microsystems, Salt Lake City, USA).The spiking signals were sampled at 20 kHz. To detect the waveforms of spikes, we band-pass filtered the signals at 250 Hz − 5 k Hz. Signals above 3.5 s.d. of the average noise level (1–5 kHz) were detected as spikes. Task-related behavioral events were digitized as TTL levels and recorded by the Cerebus system.

## Histology

The mice were deeply anesthetized with a cocktail of midazolam (7.5 mg/kg), medetomidine (0.75 mg/kg) and fentanyl (0.075 mg/kg), and perfused with paraformaldehyde (PFA, 4%). Brains were removed, fixed in 4% PFA (4°C) overnight, and then transferred to 30% sucrose in phosphate-buffered saline until equilibration. Brains were sectioned at 50–70 µm. Fluorescence images were taken with a virtual slide microscope (VS120, Olympus, Shinjuku, Japan). The atlas schematics in *Figure 4*, *Figure 4—figure supplement 2*, *Figure 4—figure supplement 6*, *Figure 4—figure supplement 7* and *Figure 5—figure supplement 1* are modified from *Franklin and Paxinos, 2007*.

To verify the location of implanted electrode, we made electrolytic lesions by applying 20 µA of current for 20 s through each channel of the electrode. Brain sections were stained with cresyl violet.

## Analysis of behavior

To estimate psychometric curves for both low-boundary and high-boundary blocks, all trials (except those within 30 trials after boundary switch) in all sessions were used to compute the correct rate for each SF. The curve of correct rate was fitted with the psychometric function using psignifit (*Wichmann and Hill, 2001*):

$$\Psi(x) = \gamma + (1 - \gamma - \lambda)\frac{1}{1+exp(-g(x))},$$
$$g(x) = \frac{x-\alpha}{\beta},$$

in which $\gamma$ and $\lambda$ represent the lower and higher asymptotes, respectively, $\alpha$ is the boundary, and $\beta$ is the slope of the curve.

To estimate the number of trials to reverse choice, we averaged the performance for reversing stimulus after boundary switch across all blocks in all sessions. We fitted an exponential function to the data of correct rate for the first 60 reversing-stimulus trials after boundary switch (*Jaramillo and Zador, 2014*):

$$f(n) = A\left(1 - e^{-n/\tau}\right) + I,$$

in which $n$ indicates trial number after a switch, $\tau$ is the speed of change, $A$ represents the asymptotic performance, and 1 - $I$ is the performance for the reversing stimulus trials just before the switch. The number of trials to reverse choice was computed as the number of trials needed to cross the 50% correct rate of the fitted curve.

The 15 trials before and the 15 trials after the switch of categorical boundary were defined as stable and switching periods, respectively. The response latency was defined as the duration between stimulus onset (200 ms after central port entry) and nose-poking the side port. We computed right choice bias to estimate the influence of a previous non-reversing stimulus on the choice for the reversing stimulus. Right choice bias following a low-frequency (high-frequency) stimulus was computed as the probability of right choice subtracted by that averaged over all reversing-stimulus trials in the low-boundary (high-boundary) block. To compute the difference in right choice bias, the right choice bias following a low-frequency stimulus was subtracted by that following a high-frequency stimulus.

## Behavioral models

To estimate the fluctuation of decision criterion on a trial-by-trial basis, and to understand the contribution of trial history to the adjustment of decision criterion, we designed a logistic regression model with a dynamic DC and fitted the model to behavioral data combined across multiple sessions. The choice in trial $t$ was modeled with a logistic function: $\mathrm{p}(t) = \frac{1}{1+e^{-z(t)}}$, where p is the probability of choosing right and z is a decision variable. For each trial $t$, z is calculated as: $z(t) = \gamma1 \times \mathrm{S}(t) - \mathrm{DC}(t)$, where S(t) is current stimulus, $\gamma1$ is the weight for current stimulus, and DC $(t)$ represents the internal decision criterion in current trial. The SF of stimulus S was normalized between $-1$ and 1, in which negative and positive values indicate lower and higher SFs, respectively, and 0 indicates the reversing stimulus. The parameter $\gamma1$ was constrained within $[0, +\infty)$. We fitted the model to choices in all trials, as well as to choices in the stable period (last 15 trials before switch) and the switching period (first 15 trials after switch) separately. For the latter case, because the categorical boundary switched once the performance for the reversing stimulus reached 70% (mice in *Figure 1*, *Figure 2* and *Figure 3*) or 80% (mice in other Figures) over the last 10 reversing-stimulus trials, the initial value of DC for the first trial in each block of stable period was set to a value corresponding to a correct rate of 70%, and the initial value of DC for the first trial in each block of switching period was set to a value corresponding to a correct rate of 30% or 20%.

Intuitively, the mice could adjust their internal DC according to the experienced action-outcome association and the previously experienced visual stimuli (*Akrami et al., 2018*; *Jaramillo and Zador, 2014*). These possibilities motivated us to design rules to update DC in a trial-by-trial manner according to specific trial history.

After a trial of reversing stimulus, DC was updated according to the following rules:

$\mathrm{DC}(t) = (1-\beta) \times \mathrm{DC}(t-1) + \alpha1$ for a rewarded left choice,
$\mathrm{DC}(t) = (1-\beta) \times \mathrm{DC}(t-1) - \alpha1$ for a rewarded right choice,
$\mathrm{DC}(t) = (1-\beta) \times \mathrm{DC}(t-1) - \alpha2$ for an unrewarded left choice,
$\mathrm{DC}(t) = (1-\beta) \times \mathrm{DC}(t-1) + \alpha2$ for an unrewarded right choice,

where $\alpha1$ and $\alpha2$ are parameters modeling the effect of rewarded and unrewarded choices, respectively, and $\beta$ (constrained within $(-\infty, 1]$) models the tendency of DC to drift towards 0 ($0 < \beta < 1$) or away from 0 ($\beta < 0$). For a positive (negative) $\alpha1$, a rewarded choice introduces a bias towards (away from) the previously chosen side, indicating a win-stay (win-shift) strategy. For a positive (negative) $\alpha2$, an unrewarded choice causes a bias away from (towards) the previously chosen side, indicating a lose-shift (lose-stay) strategy.

After a trial of non-reversing stimulus, DC was updated according to the following:

$$\mathrm{DC}(t) = (1-\beta) \times \mathrm{DC}(t-1) + \gamma2 \times \mathrm{S}(t-1),$$

where $\gamma2$ is the weight for stimulus in previous trial. Here we did not implement parameters for reward/non-reward history because the performance for non-reversing stimuli was usually much higher than that for the reversing stimulus. For $\gamma2$ with a positive value, a previously experienced low-frequency stimulus (i.e., SF lower than that of the reversing stimulus) introduces a right choice bias, while a previously experienced high-frequency stimulus (i.e., SF higher than that of the reversing stimulus) introduces a left choice bias. Note that when the previous trial is the reversing stimulus, S(t-1) becomes 0 and DC is updated according to: $\mathrm{DC}(t) = (1-\beta) \times \mathrm{DC}(t-1)$.

To evaluate whether each model parameter is necessary, we built reduced model variants, including model 2 ($\alpha1 = 0$), model 3 ($\alpha2 = 0$), model 4 ($\alpha1 = 0$ and $\alpha2 = 0$), model 5 ($\beta = 0$), model 6 ($\gamma1 = 1$) and model 7 ($\gamma2 = 0$), in addition to the full model (model 1).

Given the possibility that the mice could solve the task by updating separate value functions of different choices rather than comparing the stimulus with a decision criterion, we also designed a reinforcement learning (RL) model (*Sutton and Barto, 1998*). In this model, we used sensory stimulus in the computation of expected values of left and right choices (*Ql* and *Qr*) (*Lak et al., 2019*), and *Ql* and *Qr* are mapped into the mice's choice through a softmax function:

$$Pr(t) = \frac{e^{Qr(t)}}{e^{Ql(t)} + e^{Qr(t)}},$$

where *Pr* represents the probability of right choice. For each trial *t*, *Ql* and *Qr* are calculated as:

$$Ql(t) = \gamma \times (1 - \mathrm{S}(t)) \times Vl(t),$$

$$Qr(t) = \gamma \times \mathrm{S}(t) \times Vr(t),$$

where *Vl* and *Vr* are the value functions for left and right choices, respectively, and $\gamma$ is the weight for current stimulus S(*t*). The SF of the stimulus was normalized between 0 and 1, in which 0 and 1 correspond to the lowest and highest SFs, respectively. The model updates the value functions according to the following rules:

$Vl(t) = Vl(t-1) + \alpha \times (1 - Ql(t-1))$ for a rewarded left choice,
$Vr(t) = Vr(t-1) + \alpha \times (1 - Qr(t-1))$ for a rewarded right choice,
$Vl(t) = Vl(t-1) + \alpha \times (0 - Ql(t-1))$ for an unrewarded left choice,
$Vr(t) = Vr(t-1) + \alpha \times (0 - Qr(t-1))$ for an unrewarded right choice,

where $\alpha$ (constrained within (0, 1)) is the learning rate, and (1-*Ql*), (1-*Qr*), (0-*Ql*) or (0-*Qr*) represents reward prediction error.

For each of the models (different variants of the dynamic-DC model and the RL model), we fitted the model to behavioral data in all trials, as well as separately to choices in the stable and switching periods. The model was fitted with the method of maximum likelihood estimation, using sequential quadratic programming algorithm (*Nocedal and Wright, 2006*) to search for sets of parameters to minimize the average negative log-likelihood of the data ('*fmincon*' function in Matlab 2017b, with '*sqp*' option). We applied 200 runs of 5-fold cross-validation, with balanced number of low-boundary and high-boundary blocks in each run. The median values of the parameter distribution were reported.

We simulated behavior using the parameters of model 1 or model four fitted to the mice's actual behavior. Each simulation contained 50 low-boundary blocks and 50 high-boundary blocks, with the first block randomly chosen as low- or high-boundary block. As in the actual experiment, the boundary switched once the block consisted of at least 60 trials and the performance for the reversing stimulus reached 70% in the last 10 reversing-stimulus trials. For model simulation of choices in all trials, the DC in the first trial of a next block inherited the value in the last trial of the previous block. For model simulation of only those trials in the stable and switching periods, the DC in the first trial of the stable (switching) period was set to a value corresponding to 70% (30%) correct rate for the reversing stimulus.

To perform parameter recovery analysis, we fitted the model to simulated data by applying 200 runs of 5-fold cross-validation, and used the median values of the parameter distribution as the recovered parameters. Parameter recovery analysis was performed for the parameters of the model fitted to data in all trials, and also for those of the model fitted to choices in the stable and switching periods separately. In the latter case, we used two sets of original parameters: one was the model parameters for the 10 mice in *Figure 1*, another was a wider range of parameters, the values of which were described in the following. For model 1, the ranges of original parameters for $\alpha1$, $\alpha2$, $\beta$, $\gamma1$ and $\gamma2$ were set as [−1.5 0.8], [−0.7 1.5], [−0.2 0.9], [1 13] and [−0.4 1.5], respectively. For model 4, the ranges of original parameters for $\beta$, $\gamma1$ and $\gamma2$ were set as [−0.2 0.9], [1 13] and [−0.4 1.5], respectively. When we performed parameter recovery for one parameter (e.g., $\alpha1$ in model 1) within a wide range, the other parameters (e.g., $\alpha2$, $\beta$, $\gamma1$ and $\gamma2$ in model 1) were each fixed at a value

corresponding to the median parameters of all mice: α1, α2, β, γ1 and γ2 in model one were fixed at −0.13, 0.13, 0.22, 3.64 and 0.42, respectively, and β, γ1 and γ2 in model four were fixed at 0.21, 3.61 and 0.36, respectively.

## Analysis of neuronal responses

Spike sorting was performed offline using commercial software (Offline Sorter V3; Plexon Inc, Dallas, USA). The sorting involved an automated clustering process in 3-D principle component space of waveform features (*Shoham et al., 2003*) and a final stage of manual verification. Spike clusters were considered as single units if the interspike interval was >1 ms and the p value for multivariate ANOVA test on clusters was less than 0.05.

Except the analysis in *Figure 6C−H* and *Figure 5—figure supplement 6*, we analyzed the spikes within 0–700 ms after trial initiation (including 200 ms before stimulus presentation and 500 ms of stimulus presentation), during which the mouse held its head in the central port. For the units recorded in each session, we binned the spikes at 1 ms and computed correlation coefficients (CCs) between all pair-wise combinations of units. We also computed CC between the spike waveforms of each pair of units. Those pairs with spike CC >0.1 and spike waveform CC >0.95 were considered to be duplicate units that appeared in multiple channels, and the unit with a lower firing rate in the pair was discarded (*Zhu et al., 2015*). The same units recorded over two consecutive sessions in the same channel were tracked according to a previously reported method, which was based on quantification of the mean waveform shape, the autocorrelation, the average firing rate, and the cross-correlation with other simultaneously recorded neurons (*Fraser and Schwartz, 2012*). Only those units that were tracked for ≥2 sessions were included in the analysis.

To quantify the preference of each neuron for upcoming choice (sensory history), we applied ROC analysis (*Green and Swets, 1966*) to the distributions of spike counts on trials of different choice (different sensory history). The area under the ROC curve (auROC) indicates the accuracy with which an ideal observer can correctly classify whether a given response is recorded in one of two conditions. The ROC preference was defined as 2×(auROC – 0.5), which ranged from −1 to 1 (*Feierstein et al., 2006*). For the sliding window analysis of the correlation between left-right choice preference in current trial and stimulus preference in last trial (or in the trial before last), we used the spikes within 0–900 ms after trial initiation (including 200 ms before stimulus presentation, 500 ms of stimulus presentation and 200 ms after the 'Go' signal). The correlation coefficient was computed for each 50 ms time bin with a 25 ms moving window.

Linear decoders based on support vector machine (*Boser et al., 1992*) (SVM, *fitcsvm* function in Matlab 2017b) were trained to decode choice or sensory history from pseudopopulation of M2 neurons in response to the reversing stimulus. The decoding analysis was applied separately to four conditions: correct trials in the stable and switching periods, and wrong trials in the stable and switching periods. Spike counts of each neuron were z-score standardized. For each type of upcoming choice or sensory history, we randomly sampled 10 trials without replacement from each neuron to form a resampled dataset (*Raposo et al., 2014*). The resampling procedure was repeated for 1500 times. For each resampled dataset, we trained the SVM-based linear classifier on 75% of the resampled data, and tested it on the remaining 25% of the resampled data. This 4-fold cross-validation procedure was repeated 100 times and an averaged prediction accuracy was computed. Each condition yielded 1500 data points of prediction accuracy. To quantify the dynamics of prediction accuracy, we used the spikes within 0–900 ms after trial initiation (including 200 ms before stimulus presentation, 500 ms of stimulus presentation and 200 ms after the 'Go' signal). The SVM classifier was applied to each 50 ms time bin with a 25 ms moving window, and the resampling procedure was repeated for 100 times.

To be included in the ROC or SVM analysis, we required that a neuron should have at least 10 trials for each type of choice (sensory history) under each condition and firing rate >0.5 spikes/s in at least one of the conditions.

## Statistical analysis

No statistical methods were used to pre-determine sample sizes. Sample sizes are consistent with similar studies in the field. The statistical analysis was performed using MATLAB or GraphPad Prism (GraphPad Software). Wilcoxon two-sided signed rank test, Wilcoxon two-sided rank sum test, two-

way ANOVA or two-way repeated measures ANOVA (followed by Sidak's multiple comparisons test) was used to determine the significance of the effect. Correlation values were computed using Pearson's correlation. Unless otherwise stated, data were reported as mean ± SEM and p values < 0.05 were considered statistically significant.

## Acknowledgements

We thank Ning-long Xu and Tianming Yang for advice on the modelling of behavior and data analysis. We thank Dechen Liu for discussion on electrode implantation, Taorong Xie for the initial version of Matlab scripts of data acquisition and Yaping Li for technical assistance. This work was supported by the Strategic Priority Research Program of Chinese Academy of Sciences (grant No. XDB32010200), Shanghai Municipal Science and Technology Major Project (grant No. 2018SHZDZX05) and the National Natural Science Foundation of China (31571079, 31771151).

## Additional information

### Funding

| Funder | Grant reference number | Author |
| --- | --- | --- |
| Chinese Academy of Sciences | XDB32010200 | Haishan Yao |
| Shanghai Municipal Science and Technology Commission | 2018SHZDZX05 | Haishan Yao |
| National Natural Science Foundation of China | 31571079 | Haishan Yao |
| National Natural Science Foundation of China | 31771151 | Haishan Yao |

The funders had no role in study design, data collection and interpretation, or the decision to submit the work for publication.

### Author contributions

Tian-Yi Wang, Conceptualization, Formal analysis, Investigation, Visualization, Writing - review and editing; Jing Liu, Investigation; Haishan Yao, Conceptualization, Supervision, Funding acquisition, Writing - original draft, Project administration, Writing - review and editing

### Author ORCIDs

Tian-Yi Wang http://orcid.org/0000-0001-6488-339X
Haishan Yao https://orcid.org/0000-0003-4974-9197

### Ethics

Animal experimentation: Animal use procedures were approved by the Animal Care and Use Committee at the Institute of Neuroscience, Chinese Academy of Sciences (approval number NA-013-2019), and were in accordance with the guidelines of the Animal Advisory Committee at the Shanghai Institutes for Biological Sciences.

### Decision letter and Author response

Decision letter https://doi.org/10.7554/eLife.54474.sa1
Author response https://doi.org/10.7554/eLife.54474.sa2

## Additional files

### Supplementary files

- Transparent reporting form

## Data availability

All data generated or analyzed during this study are available on Dryad (https://doi.org/10.5061/dryad.1c59zw3rs). Source data files have been provided for Figures 1–6.

The following dataset was generated:

| Author(s) | Year | Dataset title | Dataset URL | Database and Identifier |
|---|---|---|---|---|
| Wang T-Y, Liu J, Yao H | 2020 | Data from: Control of adaptive action selection by secondary motor cortex during flexible visual categorization | https://doi.org/10.5061/dryad.1c59zw3rs | Dryad Digital Repository, 10.5061/dryad.1c59zw3rs |

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
