## [Decision Letter]

**Acceptance summary:**

This work demonstrates that chemogenetic inactivation in the secondary motor cortex (M2) impairs the performance of mice in perceptual decision making when a decision boundary is dynamically shifted but not when the task is stable, thus revealing a specific role of M2 in adaptive decision-making. This work will be of great interest to those who study the neural mechanisms of decision making.

**Decision letter after peer review:**

Thank you for submitting your article "Control of adaptive action selection by secondary motor cortex during flexible visual categorization" for consideration by *eLife*. Your article has been reviewed by three peer reviewers, and the evaluation has been overseen by a Reviewing Editor and Timothy Behrens as the Senior Editor. The following individuals involved in review of your submission have agreed to reveal their identity: Alex C Kwan (Reviewer #2); Carl CH Petersen (Reviewer #3).

The reviewers have discussed the reviews with one another and the Reviewing Editor has drafted this decision to help you prepare a revised submission.

Summary:

In this study, Wang and colleagues examined the role of secondary motor cortex (M2) in flexible decision making. Mice were trained to categorize visual stimuli based on spatial frequency, designed after the task developed by Jaramillo, Zador and their colleagues. The position of decision boundary along the spatial frequency was shifted across blocks of trials. Mice biased their choice depending on the position of decision boundary. The behavior was consistent with a model in which animals' choices were guided by the stimulus history but not by the reward history. The authors then demonstrate that chemogenetic inactivation of M2 impaired flexible changes in the animals' decision boundary although it did not impair the performance without boundary shifts.

All the reviewers thought that this study addresses an important question, uses a good task, and provides important results. However, the reviewers thought that there are various technical and interpretational concerns that need to addressed before publication of this work in *eLife*.

Essential revisions:

1) The effect of chemogenetic inactivation is rather small, and the results and the data analysis do not appear to be very robust. Given that DREADD likely results in partial inactivation, it is difficult to interpret negative results for mPFC and OFC. Although the reviewers commend that these experiments were done, the results need to be interpreted more carefully, and tests using more complete inactivation (e.g. muscimol) would be preferable.

2) The authors use model-based analysis and conclude that the animal's choices are guided by stimulus history but not by reward history. Although this is a very important effort, the reviewers identified several issues that need to be addressed.

2a) The manuscript emphasizes changes in the decision boundary (task contingency) but the model analysis indicated that the animals were not reacting to reward history but stimulus history. It seems that this is mainly due to an unusual choice of stimuli (a majority of stimuli were chosen from right next to the decision boundary) used for each block, concurrently with shifting decision boundary. This unusual choice of stimuli might have masked the effect of reward history in behavior or data analysis. This task design needs to be explained more clearly in the Results section and preferably some figures representing it. Furthermore, the motivation of this task design, as opposed to shifting decision boundary without changing the stimulus statistics, needs to be explained.

2b) The validity of model-based analysis depends on whether the model was able to fit the data reasonably well in the first place. Please provide the evidence (quantification and visualization) of goodness of fit.

2c) The authors conclude that the animals' choices were not affected by reward history based on the observation that the model that depends on stimulus history fit the data better than a reinforcement learning (RL) model. The reviewers thought that it is impossible to make such a conclusion just by a comparison with a particular RL model. The authors need to explore more thoroughly what alternative RL models may fit the data well. The current RL model that the authors used computes action values for left versus right choices without considering stimuli. A simple possibility is an RL model that computes action values specifically for each stimulus (corresponding to "states" in RL).

3) The reviewers thought that the electrophysiological recording data are not thoroughly analyzed nor presented in an informative way. The reviewers make various suggestions to improve (see below). One possibility is to remove this part altogether (Reviewer 1) but we would like to see more informative presentations and insightful analyses of the electrophysiology data.

More detailed explanations of the above points from individual reviewers are included in the following. The manuscript will be re-evaluated based on your responses to these concerns and suggestions.

Reviewer #1:

The paper by Wang et al. developed a task which requires mice to indicate whether a visual stimulus was higher or lower in spatial frequency (SF) than a boundary SF value. The boundary SF was altered between two values in two different blocks, requiring mice to adjust an internal decision criterion to obtain maximum reward.

Using a logistic regression model, the paper estimates the dependence of decisions on the stimulus, and on trial history. In doing so, it demonstrates that mouse decisions after a block switch was primarily accounted for by stimulus history (which differed between the block types) rather than the experience of errors on the stimulus condition positioned between the boundary values.

The paper demonstrates that chemogenetic inactivation in M2 impairs choice behaviour during the switching period but not during the stable period. By applying the behavioural model, the paper finds that M2 inactivation during the switch period reduces the behavioural dependence on stimulus history (for non-reversing stimuli), suggesting that M2 plays a causal role in stimulus-action remapping based on stimulus history. Interestingly, the paper shows that M2 doesn't seem to play a role during stable stimulus-choice trials, and it shows that the effect on switching trials is specific to M2 and not nearby frontal regions such as mPFC or OFC. The paper also includes results from electrophysiological recording of M2 during the task.

Overall, the behavioural experiment and the inactivation results are very interesting. Nevertheless, the electrophysiological results are hard to understand, and seem to add little to the paper. The conclusion of the paper, that M2 plays a role in flexible stimulus-choice association based on stimulus history is novel. However we have several questions and concerns:

Major concerns:

Many of the conclusions hinge on the model quality. However, there is no indication anywhere that the behavioural model is actually fitting the behavioural data well. Only comparisons between different models are presented. It is necessary and useful to visualise the model fits using psychometric curves.

The stable period is conceptualised as a period when the decision criterion is stable. Yet the model shows that the DC is affected by stimulus history and lose-shift effects (Figure 2). Thus, the stable period is not so stable by these parameters. Given this, and the fact that blocks are short (60 trials), fitting the models separately on the stable and switch period might be problematic. This is particularly the case because the paper is then performing separate model comparison for the stable and the switching period trials. As such, a better approach might be to fit models on all trials, select the best model accordingly, and fit this best model separately on different sections of the data, if necessary. Or add a new parameter to the model that can indicate stable vs. switch epochs, and fit the model once using all trials.

Related to separate model fitting and in the case of inactivation data, why not fit a model to the CNO and Saline data together, and estimate a δ parameter which estimates how much the α/β/γ parameters are changed by inactivation?

The paper relies on fitting the model separately to saline and CNO sessions, to identify specific parameters that are affected by inactivation. But the model itself could be under-constrained, meaning the parameter estimates are not stable. It would be useful to simulate data with known parameter changes, and then see if it is possible to recover those parameter changes from the model based on the number of trials that were obtained.

The comparison with the RL model: it appears that the RL model performs as good as the best regression model in the stable period but not in the switching period. What was the learning rate of the fitted RL in the switching period compared to stable period? Was there any constraints on learning rate when fitting? More generally, since the paper is considering a learning situation, the comparison with the RL model seems important and should be explained further. The class of RL model tested here can be reformulated to be analogous to the regression with dynamic decision criterion (prediction error-mediated changes in Q values can be adjusted to be analogous to changes in decision criterion). As such, it is unclear how these two models are testing competing hypotheses.

The model only allows stimulus history effects after trials of non-reversing stimuli. Surely the mouse would be adjusting the DC for stimulus history even if that stimulus was the reversing stimulus. How do the result changes if considering these stimuli?

The data shows that M2 inactivation does not affect the correct rate for non-reversing stimulus. This is surprising and interesting given many of the studies the paper cites do find robust behavioural effect of M2 inactivation across stimulus conditions (Goard et al., 2016 visual detection, Guo et al., 2015 whisker detection). In these studies, the mice are presumably in a “stable” condition regarding stimulus-choice association. Why do you think there's this discrepancy? Does this relate to using chemogenetic (here) vs. trial by trial optogenetic used in those studies?

It would be necessary and informative to see example of psychometric curves or learning curves in the inactivation condition vs. control, rather than only relying on model fits.

The results from electrophysiology experiments are cryptic and hard to follow. It might be easier and more convincing to illustrate example neurons before introducing the other analyses. For instance, Figure 5—figure supplement 2 is an interesting result and should probably be the first one mentioned for the ephys analysis. Overall, the electrophysiological experiments do not seem to add to the paper, and it might be best to be removed from the paper.

Related to electrophysiological data, it is hard to understand the need to use three different analysis methods: regression, ANOVA, and ROC analysis, each doing slightly different things.

There are several instances of statistical tests not correcting for multiple comparisons. For instance, in Figure 4D, the effect of M2 inactivation on the percent correct for reversing stimuli seems to be statistically significant primarily due to data from 4 mice. Does this effect stay after correcting for multiple comparisons?

Reviewer #2:

This paper by Tian-Yi Wang and colleagues describes a series of experiments to study the role of mouse M2 in adaptive action selection. The strength of this paper is the rigor. The experiments were based on a well-designed task involving flexible stimulus categorization (that have been pioneered in rodents by Zador, Jaramillo, et al.). The authors also did a great job putting together a computational model that provides considerable insights into the mouse's behavioral strategy. This led to an intriguing behavioral conclusion that mice are doing the task by using sensory history but not reward-based learning.

In terms of the neural conclusion that M2 is involved in adaptive sensory-motor selection, there are a few other studies now suggesting that M2 is involved in driving sensory cue-guided choices following a switch in contingencies. Nevertheless, there is still substantial value here because the study is excellent and provides arguably the strongest evidence to date. There is also additional conceptual novelty in looking at region differences, comparing M2 with mPFC and OFC.

The manuscript is well-written, and very easy to follow and understand.

Overall, the study is technically sound and conceptually important.

Major comments:

– The neural activity analysis, correlating ROC selectivity value for previous stimulus preference (non-reversing stimulus trial) and current left-right choice preference (reversing stimulus trial) (Figure 5D), is taken as evidence that M2 neurons use sensory history to influence current choice. The analyses were done for a particular time window of a trial. What happens if this analysis was applied to a sliding window starting from last trial to current or even next trial? When does this sensory-choice coupling emerge and when does it end? This is different from the decoding analysis is Figure 6, because it speaks to the interaction rather than decoding of choice or stimulus alone.

– Again, because Figure 5D is important – currently this analysis was done for cases when current trial was the reversing stimulus and the prior trial was the non-reversing stimulus. What about for other trial conditions? Do we still see the correlation in the sensory and motor related neural signals? In particular, what about the case when the current trial was the reversing stimulus and the prior trial was also a reversing stimulus?

– The comparison between M2 and mPFC and OFC is important. The results were presented as Figure 4—figure supplement 5 for mPFC and figure supplement 6 for OFC. I feel that these are exciting results demonstrating regional differences. At least some parts of each should be moved to be a main figure.

Reviewer #3:

Wang, Liu and Yao study the role of M2 in mice during a visual categorization task. Mice were trained to obtain water reward on left vs. right depending upon the spatial frequency of a visual grating with a variable decision boundary. Through modeling of decision criteria, chemogenetic inactivation and electrophysiological recordings, the authors conclude that M2 contributes to flexible stimulus-action coupling.

I think the behavior is well-designed and mice seem to perform well. I also like the quantitative modeling of the behavior.

1) I find the overall effect of the DREADD inactivation of M2 on behavior to be small. It is not obvious to me that DREADD inactivation is being applied in a useful way here. Given that there is no cell-type-specific manipulation, it would probably have been simpler and better to use pharmacological inactivation (e.g. muscimol). This would likely give a complete inactivation of M2 rather than the reduction to ~30% activity currently shown in Figure 4B. Perhaps larger effects upon behavior might have been observed. The small effect size reported for M2, also means that the negative effects for mPFC and OFC inactivation are less impressive, although it is very good that the authors carried out these further experiments.

2) The electrophysiological data are summarized in Figure 5 as correlations, but the overall description of the data is rather limited. I think the authors could give a more extended analysis of spiking activity across trial time, including showing example neurons. I imagine that similar effects might be found in multiple other brain regions, if they were recorded.

3) I am somewhat concerned by the choice of stimuli presented to the mice. I read that the type of visual stimulus depends upon the boundary frequency. For example: "For the low boundary block, gratings at 0.03 and 0.095 cycles/^o^ were presented for 90% of trials, gratings at the other frequencies were presented for 10% of trials, and the boundary frequency was between 0.065 and 0.095 cycles/^o^. For the high-boundary block, gratings at 0.095 and 0.3 cycles/^o^ were presented for 90% of trials, and the boundary frequency was between 0.095 and 0.139 cycles/^o^." I think the statistics of presented stimuli will change perceptual thresholds. Why not use the same stimulus set throughout? This would seem to be fairer.

---

## [Author Response]

Essential points:1) The effect of chemogenetic inactivation is rather small, and the results and the data analysis do not appear to be very robust. Given that DREADD likely results in partial inactivation, it is difficult to interpret negative results for mPFC and OFC. Although the reviewers commend that these experiments were done, the results need to be interpreted more carefully, and tests using more complete inactivation (e.g. muscimol) would be preferable.

We thank the reviewers for the suggestion. We agree with the reviewers that muscimol inactivation of M2 may produce larger effect. However, due to the reason explained below, we chose to use the DREADD inactivation method.

In our experiment, most mice made less than 7 switches per session (3.94 ± 0.83 (mean ± SD) switches/session in CNO sessions and 4.07 ± 0.69 (mean ± SD) switches/session in saline sessions for M2 inactivation experiment), and we measured 8.7 ± 0.95 (mean ± SD) saline sessions and 8.7 ± 0.95 (mean ± SD) CNO sessions for each mouse to obtain enough number of trials for psychometric curve analysis and modelling analysis. Using the method of pharmacological inactivation would require that injection of muscimol/saline with micropipette be performed for about 18 sessions for each mouse. Such large numbers of micropipette penetration may itself cause brain tissue damage. Therefore, instead of the pharmacological method, we used the DREADD inactivation method, in which CNO or saline was injected intraperitoneally.

We have added the number of CNO and saline sessions in the Materials and methods of the revised manuscript. We also mentioned in the Discussion that “it should be noted that our chemogenetic manipulation did not produce a complete inactivation of neuronal activity and thus the negative effect of mPFC/OFC inactivation may need to be further confirmed using experiments with more complete inactivation”.

2) The authors use model-based analysis and conclude that the animal's choices are guided by stimulus history but not by reward history. Although this is a very important effort, the reviewers identified several issues that need to be addressed.2a) The manuscript emphasizes changes in the decision boundary (task contingency) but the model analysis indicated that the animals were not reacting to reward history but stimulus history. It seems that this is mainly due to an unusual choice of stimuli (a majority of stimuli were chosen from right next to the decision boundary) used for each block, concurrently with shifting decision boundary. This unusual choice of stimuli might have masked the effect of reward history in behavior or data analysis. This task design needs to be explained more clearly in the Results section and preferably some figures representing it. Furthermore, the motivation of this task design, as opposed to shifting decision boundary without changing the stimulus statistics, needs to be explained.

We have explained the task and stimuli more clearly in the revised manuscript. Our visual task is similar to the auditory flexible categorization task described in a previous study (Jaramillo et al., 2014). For the visual system, the perceived size (or SF) of a visual object changes with viewing distance, and categorization of a visual stimulus as low or high SF may be adaptive to the change of viewing distance. Consistent with such change, the stimulus statistics differed between the low-boundary and high-boundary blocks in our task. This has been added in the revised manuscript.

2b) The validity of model-based analysis depends on whether the model was able to fit the data reasonably well in the first place. Please provide the evidence (quantification and visualization) of goodness of fit.

In the revised manuscript, we have provided visualization of model simulation and performed parameter recovery analysis.

2c) The authors conclude that the animals' choices were not affected by reward history based on the observation that the model that depends on stimulus history fit the data better than a reinforcement learning (RL) model. The reviewers thought that it is impossible to make such a conclusion just by a comparison with a particular RL model. The authors need to explore more thoroughly what alternative RL models may fit the data well. The current RL model that the authors used computes action values for left versus right choices without considering stimuli. A simple possibility is an RL model that computes action values specifically for each stimulus (corresponding to "states" in RL).

We have used a RL model in which sensory stimuli were used in the computation of action values (Lak et al., 2019). In this RL model, the expected value of left (right) choice is the learned value of left (right) choice weighted by current sensory stimulus, with the parameters α and γ representing learning rate and stimulus weight, respectively.

Since stimuli at adjacent SFs were similar and some stimuli were presented with a low probability, it is natural for us to assume the form in which action values are weighted by current sensory stimulus, rather than to compute action values specifically for each stimulus.

Combining the result of model comparison and the histogram of the number of mice best fit by each model (7 variants of the dynamic-DC model and the RL model), we found that model 4 of the dynamic-DC model (α1 = 0 and α2 = 0) was the winning model (Figure 3 and Figure 3—figure supplement 1).

3) The reviewers thought that the electrophysiological recording data are not thoroughly analyzed nor presented in an informative way. The reviewers make various suggestions to improve (see below). One possibility is to remove this part altogether (Reviewer 1) but we would like to see more informative presentations and insightful analyses of the electrophysiology data.

We have removed the regression and ANOVA analysis of the electrophysiological recording data. However, we believe that the analysis on the choice preference and the previous-stimulus preference is important, as we found that the preference for left-right choice in current trial was intimately related to the preference for stimulus in last trial, consistent with the finding that sensory history was important for correct action selection in the task. We also provided more informative presentations of M2 activity and additional sliding window analysis on the correlation between choice preference and previous-stimulus preference.

More detailed explanations of the above points from individual reviewers are included in the following. The manuscript will be re-evaluated based on your responses to these concerns and suggestions.Reviewer #1:[…]Major concerns:Many of the conclusions hinge on the model quality. However, there is no indication anywhere that the behavioural model is actually fitting the behavioural data well. Only comparisons between different models are presented. It is necessary and useful to visualise the model fits using psychometric curves.

We have provided visualization of the simulated behavior for our models and performed parameter recovery analysis, which are described in the responses below.

The stable period is conceptualised as a period when the decision criterion is stable. Yet the model shows that the DC is affected by stimulus history and lose-shift effects (Figure 2). Thus, the stable period is not so stable by these parameters. Given this, and the fact that blocks are short (60 trials), fitting the models separately on the stable and switch period might be problematic. This is particularly the case because the paper is then performing separate model comparison for the stable and the switching period trials. As such, a better approach might be to fit models on all trials, select the best model accordingly, and fit this best model separately on different sections of the data, if necessary. Or add a new parameter to the model that can indicate stable vs. switch epochs, and fit the model once using all trials.

We thank the reviewer for raising this point. In addition to fitting the models separately to choices in the stable and switch periods (Figure 2), we have used each of the models to fit choices in all trials (Figure 3—figure supplement 1). Combining the result of model comparison and the histogram of the number of mice best fit by each model, we found that model 4 of the dynamic-DC model (α1 = 0 and α2 = 0) remained the winning model (Figure 3—figure supplement 1).

We also visualized the simulated behavior for model 1 and model 4, using the parameters of the model fitted to data in all trials. As shown by Figure 3—figure supplement 2, both models could capture the dynamic change in performance for the reversing stimulus after the boundary switch. The number of trials to reverse choice estimated from model 1 simulation tended to be larger than that estimated from the actual performance, whereas that estimated from model 4 simulation matched well with the actual data (Figure 3—figure supplement 2).

Related to separate model fitting and in the case of inactivation data, why not fit a model to the CNO and Saline data together, and estimate a δ parameter which estimates how much the α/β/γ parameters are changed by inactivation?

As described above, we fitted each of the models to choices in all trials and performed model comparison analysis. We found that model 4 of the dynamic-DC model (α1 = 0 and α2 = 0) was the winning model (Figure 3—figure supplement 1).

We also performed parameter recovery analysis to check that our fitting procedure can accurately estimate parameters. We simulated model 1 and model 4, respectively, using the parameters of the model fitted to data in all trials for the 10 mice in Figure 1. For both models, the recovered parameters matched the original parameters (Figure 3—figure supplement 4). We also simulated model 1 or model 4 using the parameters of the model fitted separately to choices in the stable and switching periods. This was performed using two sets of parameters: one was the model parameters for the 10 mice in Figure 1, another was a wider range of parameters (see Materials and methods in the revised manuscript). For both cases, there was good agreement between the recovered and original parameters (Figure 3—figure supplement 5 and Figure 3—figure supplement 6).

The above analysis established that both model 1 and model 4 could recover original parameters from simulated data, we thus preferred to fit the model separately to CNO and saline data, and compare the model parameters between the two conditions.

The paper relies on fitting the model separately to saline and CNO sessions, to identify specific parameters that are affected by inactivation. But the model itself could be under-constrained, meaning the parameter estimates are not stable. It would be useful to simulate data with known parameter changes, and then see if it is possible to recover those parameter changes from the model based on the number of trials that were obtained.

We thank the reviewer for this suggestion. Using the parameter recovery analysis, we found that the recovered sensory history parameter γ2 in the switching period was significantly lower in CNO than in saline sessions, whereas the recovered parameter γ2 in the stable period was not significantly different between CNO and saline sessions, consistent with the actual parameter changes (Figure 4—figure supplement 5).

The comparison with the RL model: it appears that the RL model performs as good as the best regression model in the stable period but not in the switching period. What was the learning rate of the fitted RL in the switching period compared to stable period? Was there any constraints on learning rate when fitting? More generally, since the paper is considering a learning situation, the comparison with the RL model seems important and should be explained further. The class of RL model tested here can be reformulated to be analogous to the regression with dynamic decision criterion (prediction error-mediated changes in Q values can be adjusted to be analogous to changes in decision criterion). As such, it is unclear how these two models are testing competing hypotheses.

We thank the reviewer for raising this point. We agree with the reviewer that prediction error-mediated changes in Q values are somehow analogous to changes in decision criterion. However, the contribution of outcome history and sensory history can be separately analyzed in the dynamic-DC model, whereas the two factors were less separable in the RL model. We have revised our statement to: “we used the RL model to test an alternative hypothesis that the mouse might be updating the value functions of left and right choices separately, rather than comparing the current stimulus to one single decision boundary”.

In the RL model, the expected value of left (right) choice is the learned value of left (right) choice weighted by current sensory stimulus (Lak et al., 2019). The expected values of left and right choices (*Ql* and *Qr*) are mapped into the mice’s choice through a softmax function:Pr(t)=eQr(t)eQl(t)+eQr(t),where *Pr* represents the probability of right choice. For each trial *t*, *Ql* and *Qr* are calculated as:Ql(t)=γ×(1−S(t))×Vl(t),Qr(t)=γ×S(t)×Vr(t),where *Vl* and *Vr* are the value functions for left and right choices, respectively, and γ is the weight for current stimulus S(*t*). The SF of the stimulus was normalized between 0 and 1, in which 0 and 1 correspond to the lowest and highest SFs, respectively. The model updates the value functions according to the following rules:

Vl(t)=Vl(t−1)+α×(1−Ql(t−1)) for a rewarded left choice,

Vr(t)=Vr(t−1)+α×(1−Qr(t−1)) for a rewarded right choice,

Vl(t)=Vl(t−1)+α×(0−Ql(t−1)) for an unrewarded left choice,

Vr(t)=Vr(t−1)+α×(0−Qr(t−1)) for an unrewarded right choice,

where α is the learning rate, which is constrained within [0 1] (0 < α < 1), and (1-*Ql*), (1-*Qr*), (0-*Ql*) or (0-*Qr*) represents reward prediction error.

For the RL model fitted to choices in the stable and switching periods separately, the learning rate (α) in the switching period was significantly higher than that in the stable period (p = 0.002, Wilcoxon signed rank test, Figure 3).

We also used each of the models (the RL model and 7 variants of the dynamic-DC model) to fit choices in all trials (Figure 3—figure supplement 1). We found that the CV likelihood of model 1 or model 4 was both significantly higher than that of the RL model (Figure 3—figure supplement 1E and F). Combining the result of model comparison and the histogram of the number of mice best fit by each model, we found that model 4 of the dynamic-DC model (α1 = 0 and α2 = 0) was the winning model (Figure 3—figure supplement 1D−G).

The model only allows stimulus history effects after trials of non-reversing stimuli. Surely the mouse would be adjusting the DC for stimulus history even if that stimulus was the reversing stimulus. How do the result changes if considering these stimuli?

In our model, after a trial of non-reversing stimulus, DC was updated according to the following: DC(t)=(1−β)×DC(t−1)+γ2×S(t−1), where γ2 is the weight for stimulus in previous trial. The S was normalized between -1 and 1, in which negative and positive values indicate lower and higher SFs, respectively, and 0 indicates the reversing stimulus. When the previous trial is the reversing stimulus, S(t-1) becomes 0 and DC is updated according to: DC(t)=(1−β)×DC(t−1). This has been clarified in the revised manuscript.

The data shows that M2 inactivation does not affect the correct rate for non-reversing stimulus. This is surprising and interesting given many of the studies the paper cites do find robust behavioural effect of M2 inactivation across stimulus conditions (Goard et al., 2016 visual detection, Guo et al., 2015 whisker detection). In these studies, the mice are presumably in a “stable” condition regarding stimulus-choice association. Why do you think there's this discrepancy? Does this relate to using chemogenetic (here) vs. trial by trial optogenetic used in those studies?

A recent study found that bilateral muscimol inactivation of M2 slowed the transition from action-guided to sound-guided response, without affecting the high performance of sound-guided trials after the transition (Siniscalchi et al., 2016). Consistent with this study, we found that bilateral inactivation of M2 slowed the adaptive adjustment of action selection following the boundary switch, without affecting choice behavior in steady state.

In our study, the visual stimulus was presented until the mouse chose one of the two side ports. However, the task in Goard et al., 2016 and Guo et al., 2014, both involved a short-term memory component. In the study of Guo et al., 2014, unilateral optogenetic inactivation of ALM caused an ipsilateral bias, with a stronger effect of inactivation during the delay epoch than the sample epoch. Erlich et al., 2011, found that unilateral inactivation of the rat FOF (equivalent to M2) generated a contralateral impairment, with stronger effect in memory than in non-memory trials. As our task did not involve a memory component, our result that the performance for non-reversing stimulus was not affected by M2 inactivation is consistent with the smaller effect of M2 inactivation on non-memory than on memory trials (Erlich et al., 2011; Goard et al., 2016; Guo et al., 2014).

It would be necessary and informative to see example of psychometric curves or learning curves in the inactivation condition vs. control, rather than only relying on model fits.

We have plotted psychometric curves and example performance curves for reversing stimulus in CNO vs. saline conditions (Figure 4—figure supplement 1).

The results from electrophysiology experiments are cryptic and hard to follow. It might be easier and more convincing to illustrate example neurons before introducing the other analyses. For instance, Figure 5—figure supplement 2 is an interesting result and should probably be the first one mentioned for the ephys analysis. Overall, the electrophysiological experiments do not seem to add to the paper, and it might be best to be removed from the paper.Related to electrophysiological data, it is hard to understand the need to use three different analysis methods: regression, ANOVA, and ROC analysis, each doing slightly different things.

We have removed the regression and ANOVA analysis of the electrophysiological recording data. However, we believe that the analysis on the choice preference and the previous-stimulus preference is important, as we found that the preference for left-right choice in current trial was intimately related to the preference for stimulus in last trial, consistent with the finding that sensory history was important for correct action selection in the task. We also found that the representations of upcoming choice and sensory history in M2 were stronger during the switching than the stable period, which may account for the important role of M2 in adaptive action selection in the switching period.

To facilitate understanding of the analysis, we provided spike rasters and PSTHs of example M2 neurons in different choice conditions and in different sensory-history conditions (Figure 5A-D) before introducing the other analyses.

There are several instances of statistical tests not correcting for multiple comparisons. For instance, in Figure 4D, the effect of M2 inactivation on the percent correct for reversing stimuli seems to be statistically significant primarily due to data from 4 mice. Does this effect stay after correcting for multiple comparisons?

For the effect of M2 inactivation shown in Figure 4C and D, we have performed two-way ANOVA followed by Sidak’s multiple comparisons test, which showed that the reduction in performance for reversing stimulus was significant in the switching period (p = 0.022). This has been clarified in the revised manuscript.

Reviewer #2:[…]Major comments:– The neural activity analysis, correlating ROC selectivity value for previous stimulus preference (non-reversing stimulus trial) and current left-right choice preference (reversing stimulus trial) (Figure 5D), is taken as evidence that M2 neurons use sensory history to influence current choice. The analyses were done for a particular time window of a trial. What happens if this analysis was applied to a sliding window starting from last trial to current or even next trial? When does this sensory-choice coupling emerge and when does it end? This is different from the decoding analysis is Figure 6, because it speaks to the interaction rather than decoding of choice or stimulus alone.

We thank the reviewer for this suggestion. We computed the correlation coefficient between the preference for left-right choice in current trial and the preference for stimulus in the last trial using a sliding window of 50 ms (Figure 5—figure supplement 6A and B). We found significant negative correlation for all time bins, including those time bins between central port entry and stimulus onset. For the correlation between the preference for left-right choice in current trial and the preference for stimulus in the trial before last (Figure 5—figure supplement 6C and D), the statistical significance of the correlation diminished in most time bins. Thus, the result suggests that the choice preference of M2 neurons in current trial is strongly influenced by stimulus history, and the preference for choice in current trial is coupled to the preference for stimulus in the last but not earlier trial.

– Again, because Figure 5D is important – currently this analysis was done for cases when current trial was the reversing stimulus and the prior trial was the non-reversing stimulus. What about for other trial conditions? Do we still see the correlation in the sensory and motor related neural signals? In particular, what about the case when the current trial was the reversing stimulus and the prior trial was also a reversing stimulus?

In Figure 5D (now in Figure 5L in the revised manuscript), we analyzed the relationship between the preference for choice in current reversing-stimulus trial and the preference for stimulus in previous trial. To compute the preference for previous stimulus, we divided the responses of current trial to two groups according to the SF in previous trial (previous low-frequency vs. previous high-frequency). When the previous trial was also the reversing stimulus, we were not able to divide the responses to two groups to compute the previous-stimulus preference.

For sensory-choice coupling in other trial conditions, we may also consider the case when the current trial was a non-reversing stimulus. Because the non-reversing stimulus at the lowest SF and the reversing stimulus were presented for 90% of trials in the low-boundary block, and the non-reversing stimulus at the highest SF and the reversing stimulus were presented for 90% of trials in the high-boundary block, we could consider analyzing the responses for the non-reversing stimulus at the lowest or the highest SF. In such case, however, we were not able to divide the responses of current trial to two groups according to the SF in previous trial, because all previous trials would have a SF higher (or lower) than that in current trial.

Therefore, we analyzed sensory-choice coupling only when the current trial was the reversing stimulus and the previous trial was the non-reversing stimulus.

– The comparison between M2 and mPFC and OFC is important. The results were presented as Figure 4—figure supplement 5 for mPFC and figure supplement 6 for OFC. I feel that these are exciting results demonstrating regional differences. At least some parts of each should be moved to be a main figure.

As reviewer 3 pointed out that “the small effect size reported for M2 means that the negative effects for mPFC and OFC inactivation are less impressive”, we thus chose to keep the results of mPFC and OFC inactivation in figure supplements.

To address the concern of reviewer 3, we also mentioned in the Discussion that “it should be noted that our chemogenetic manipulation did not produce a complete inactivation of neuronal activity and thus the negative effect of mPFC/OFC inactivation may need to be further confirmed using experiments with more complete inactivation”.

Reviewer #3:Wang, Liu and Yao study the role of M2 in mice during a visual categorization task. Mice were trained to obtain water reward on left vs. right depending upon the spatial frequency of a visual grating with a variable decision boundary. Through modeling of decision criteria, chemogenetic inactivation and electrophysiological recordings, the authors conclude that M2 contributes to flexible stimulus-action coupling.I think the behavior is well-designed and mice seem to perform well. I also like the quantitative modeling of the behavior.1) I find the overall effect of the DREADD inactivation of M2 on behavior to be small. It is not obvious to me that DREADD inactivation is being applied in a useful way here. Given that there is no cell-type-specific manipulation, it would probably have been simpler and better to use pharmacological inactivation (e.g. muscimol). This would likely give a complete inactivation of M2 rather than the reduction to ~30% activity currently shown in Figure 4B. Perhaps larger effects upon behavior might have been observed. The small effect size reported for M2, also means that the negative effects for mPFC and OFC inactivation are less impressive, although it is very good that the authors carried out these further experiments.

We thank the reviewer for pointing out this issue. We agree with the reviewer that muscimol inactivation of M2 may produce larger effect. However, due to the reason explained below, we chose to use the DREADD inactivation method.

In our experiment, most mice made less than 7 switches per session (3.94 ± 0.83 (mean ± SD) switches/session in CNO sessions and 4.07 ± 0.69 (mean ± SD) switches/session in saline sessions for M2 inactivation experiment), and we measured 8.7 ± 0.95 (mean ± SD) saline sessions and 8.7 ± 0.95 (mean ± SD) CNO sessions for each mouse to obtain enough number of trials for psychometric curve analysis and modelling analysis. Using the method of pharmacological inactivation would require that injection of muscimol/saline with micropipette be performed for about 18 sessions for each mouse. Such large numbers of micropipette penetration may itself cause brain tissue damage. Therefore, instead of the pharmacological method, we used the DREADD inactivation method, in which CNO or saline was injected intraperitoneally.

We have added the number of CNO and saline sessions in the Materials and methods of the revised manuscript. We have mentioned in the Discussion the limitation of DREADD inactivation: “it should be noted that our chemogenetic manipulation did not produce a complete inactivation of neuronal activity and thus the negative effect of mPFC/OFC inactivation may need to be further confirmed using experiments with more complete inactivation”.

2) The electrophysiological data are summarized in Figure 5 as correlations, but the overall description of the data is rather limited. I think the authors could give a more extended analysis of spiking activity across trial time, including showing example neurons. I imagine that similar effects might be found in multiple other brain regions, if they were recorded.

We thank the reviewer for this suggestion. We have added spike rasters and PSTHs of example M2 neurons, with the responses grouped by choice or by previous stimulus (Figure 5A-D).

We also performed a sliding window analysis of correlation between the preference for left-right choice in current trial and the preference for stimulus in last trial (or the trial before last) (Figure 5—figure supplement 6).

As our electrophysiological recordings were performed from M2, we do not know whether similar effects are found from other brain regions.

3) I am somewhat concerned by the choice of stimuli presented to the mice. I read that the type of visual stimulus depends upon the boundary frequency. For example: "For the low boundary block, gratings at 0.03 and 0.095 cycles/^o^ were presented for 90% of trials, gratings at the other frequencies were presented for 10% of trials, and the boundary frequency was between 0.065 and 0.095 cycles/^o^. For the high-boundary block, gratings at 0.095 and 0.3 cycles/^o^ were presented for 90% of trials, and the boundary frequency was between 0.095 and 0.139 cycles/^o^." I think the statistics of presented stimuli will change perceptual thresholds. Why not use the same stimulus set throughout? This would seem to be fairer.

Our visual task is similar to the auditory flexible categorization task described in a previous study (Jaramillo et al., 2014). For the visual system, the perceived size (or SF) of a visual object changes with viewing distance, and categorization of a visual stimulus as low or high SF may be adaptive to the change of viewing distance. Consistent with such change, the stimulus statistics differed between the low-boundary and high-boundary blocks in our task.

We found that mice depended on sensory history to correctly change stimulus-action association when categorical boundary switched between blocks. As the switch in categorization boundary in our task was accompanied by a change in stimulus statistics, this may promote a strategy that involves sensory history.

We have clarified the above issue in the revised manuscript.